



# Spectral albedo measurements over snow-covered slopes: theory and slope effect corrections

Ghislain Picard[1], Marie Dumont[2], Maxim Lamare[1,2], Francois Tuzet[1,2], Fanny Larue[1], Roberta Pirazzini[3], and Laurent Arnaud[1]

[1]Univ. Grenoble Alpes, CNRS, Institut des Géosciences de l'Environnement (IGE), UMR 5001, Grenoble, F-38041, France
[2]Univ. Grenoble Alpes, Université de Toulouse, Météo-France, CNRS, CNRM, Centre d'Etudes de la Neige, 38000 Grenoble, France
[3]Finnish Meteorological Institute, Helsinki, Finland

**Correspondence:** Ghislain Picard (ghislain.picard@univ-grenoble-alpes.fr)

**Abstract.** Surface albedo is an essential variable to determine the Earth's surface energy budget, in particular for snow-covered areas where it is involved in one of the most powerful positive feedback loops of the climate system. Measurements of broadband albedo are therefore common in meteorology. Measurements of spectral albedo are less frequent but provide richer information, useful to understand the physical and chemical properties driving albedo variations. Both types of measurements are subject to several artefacts. Here we investigate the sensitivity of spectral albedo measurements to surface slope, and pro-

pose simple correction algorithms to retrieve the intrinsic albedo of a slope from measurements, as if it were flat. For this, we first derive the analytical equations relating albedo measured on a slope to intrinsic direct and diffuse albedo, the apportionment between diffuse and direct incoming radiation, and slope inclination and aspect. The theory accounts for two main slope effects. First, the slope affects the proportion of solar radiation intercepted by the surface relative to that intercepted by the upward-looking, horizontal, sensor. Second, the upward and downward looking sensors receive reduced radiation from the sky

and the surface respectively, and increased radiation from neighbouring terrain. Using this theory, we show that i) slope has a significant effect on albedo (over 0.01) from as little as a $\approx 1°$ inclination, causing distortions of the albedo spectral shape, ii) the first order slope effect is sufficient to fully explain measured albedo up to $\approx 15°$, which we designate as "small slope approximation", and iii) for larger slopes, the theory depends on the neighbouring slope geometry and land cover, leading to much more complex equations. Next, we derive four correction methods from the small slope approximation, to be used de-

pending on whether 1) the slope inclination and orientation are known or not, 2) the snow surface is free of impurities or dirty and 3) a single or a time-series of albedo measurements is available. The methods applied to observations taken in the Alps on terrain with up to nearly 20° slopes, prove the ability to recover intrinsic albedo with a typical accuracy of 0.03 or better. From this study, we draw two main recommendations for future field campaigns: first, sloping terrain requires more attention

because it reduces the measurement accuracy of albedo even for barely invisible slopes (1-2°). Second, while the correction of the slope effect is possible, it requires additional information such as the spectral diffuse and direction apportionment, and if possible the actual slope inclination and aspect especially when the absence of impurities can not be assumed.





# 1  Introduction

The solar irradiance absorbed by snow-covered surfaces (or net shortwave flux $\Delta SW$) is an important term of the surface energy budget that drives surface temperature and snow melt. This term is usually computed from the surface broadband albedo (the percent of radiation reflected by the surface, $\alpha$, between 300 and 5000 nm) and the incoming broadband irradiance

on the surface $E^\downarrow$, relying on the energy conservation principle to relate absorption and reflection, such as $\Delta SW = (1-\alpha)E^\downarrow$. For this reason, broadband albedo is a common meteorological variable measured using two horizontal radiation sensors, one looking upward and the other looking downward (Driemel et al., 2018; van den Broeke, 2004). Despite an apparent simplicity, measuring albedo is a notoriously difficult problem (Michalsky and Hodges, 2013) and the equation hides several potential caveats related to the angular and spectral distribution of the incoming irradiance and snow reflectance (e.g. Lee et al., 2011;

Wang and Zender, 2010). Here we focus on one of these difficulties, arising when the terrain is not flat and the sensors are consequently not parallel to the surface. In such a case, the albedo computed as the ratio of the readings from the two sensors (hereinafter called measured or apparent albedo) differs from the intrinsic surface albedo $\alpha$ (that of the surface if it were flat, also called true albedo by some authors) needed to compute $\Delta SW$. This issue concerns most snow fields in practice because it affects slopes as small as $2°$ (Grenfell et al., 1994; Larue et al., submitted). As a first consequence of the slope, albedo values

over 1 may be measured in the case of sun-facing slopes (Grenfell et al., 1994), even with perfect instruments, highlighting that measured albedo is not a well-defined reflectance bounded between 0 and 1 as it should, to respect the energy conservation principle. Another consequence visible when acquiring albedo time-series with a sub-daily resolution,is a spurious daily cycle due to variations of the local solar zenith angle on the slope during the course of the sun in the sky, even if the surface properties have not changed (Weiser et al., 2016). Such cycles can be mistakenly interpreted as a diurnal change of surface snow properties

(e.g. snow specific surface area) if the slope is not considered. Such situations occur even in the Antarctic and Greenland interior where despite an extremely flat surface at large scales, the local slope in the footprint can be significant because of sastrugi or dunes (Grenfell et al., 1994; Warren et al., 1998; Pirazzini, 2004; Wang and Zender, 2010). These two consequences lead to visible effects in some cases that should raise observer's attention during data quality checks. However in other cases the slope effect is present but barely visible. For instance, albedo on slopes facing away from the sun is systematically lower than 1 and

may look like a normal flat-surface albedo if the slope is small (Grenfell et al., 1994). Similarly cross-calibration errors or an imperfect angular response of the light collector can compensate the effect of small slopes.

A potential solution to overcome the effect of the slope and obtain intrinsic albedo is to set the sensors parallel to the terrain (Wu et al., 2018). However, it seems unpractical because the accuracy of the parallelism between the sensors and the terrain required to remove all spurious daily variations is of the order of 0.1–0.2° (Picard et al., 2016b). Such an accuracy is

unreachable, at least because the measurement accuracy of slope is rarely better than $1°$ in practice (Larue et al., submitted). Moreover the slope of the surface is often changing over a season, due to snow redistribution by wind and melt, preventing unattended albedo monitoring with such a parallel setting. Because measuring albedo with horizontal sensors is practical, and because numerous long time-series exist (Driemel et al., 2018; Fausto and van As, 2019), algorithms correcting for the slope have been developed. Weiser et al. (2016) proposed a correction method exploiting the diurnal cycle of the upwelling light



flux. Nevertheless, the study neglected the zenith angular dependence of snow albedo (Warren and Wiscombe, 1980), which induces a natural diurnal cycle that is intertwined with the spurious cycle caused by the slope. Untangling the natural and spurious cycles is required to avoid an over-correction. Despite this limitation, their method accounts for both terrain slope and sensor tilt. Tilt is another frequent issue with albedo measurements and is related to the slope problem, at first order. Despite

being beyond the scope of the present study, it is worth citing Bogren et al. (2016) who investigated the sensitivity of albedo measurements to sensor tilt through simple modelling, and Wang et al. (2016) who proposed a correction method exploiting the diurnal cycle. All these studies target broadband albedo.

Spectral albedo measurements are less frequent than broadband albedo, but provide richer information, enabling not only to establish the shortwave radiative budget, but also to investigate if and how surface albedo is driven by snow microstructural

properties (Gallet et al., 2011; Carmagnola et al., 2013; Libois et al., 2015; Carlsen et al., 2017), liquid water (Dumont et al., 2017), impurities (Skiles et al., 2018; Tuzet et al., 2017), or algea (Painter et al., 2001). As is the case with broadband albedo, spectral albedo measurements are affected by slope. In the range 400-600 nm where the intrinsic snow albedo is close to 1, a small slope is sufficient to raise the measured value above 1 (Grenfell et al., 1994; Wuttke et al., 2006). Correction methods for spectral albedo can be more elaborate than for broadband albedo, by exploiting the richness of the spectral information. Dumont

et al. (2017) propose a method to jointly correct slope and estimate snow specific surface area and impurity concentration. The method relies on the diurnal cycle similarly to Weiser et al. (2016), but accounts for the natural dependence to the incident angle at the cost of requiring extra information to separate the direct (sun) and diffuse components (e.g. spectral measurements of the diffuse-to-total ratio). Their method also assumes a particular theoretical form for the albedo, which is available and accurate for a thick layer of dry and pristine snow (e.g. Kokhanovsky and Zege, 2004) but is more problematic for wet, dirty

or shallow snow (Tuzet et al., 2019). In the present paper, we propose several complementary methods requiring either weaker assumptions or that apply to single acquisitions instead of time-series.

To develop slope correction methods for spectral albedo, the first step is to establish the equations linking measured albedo to intrinsic albedo accounting for the slope and illumination conditions. Mathematically, this problem is closely related to the widely-addressed problems of the distribution of the solar radiation at the surface and remote sensing data correction in

mountainous areas (e.g. Dozier, 1980; Lenot et al., 2009; Lee et al., 2011). The calculation of the direct (sun) component is a trivial geometrical problem once the slope and the cast and self shadows are known (Grenfell et al., 1994; Dozier and Frew, 1990). The diffuse component is more complex because it includes several contributions (the sky, surrounding terrain, and multiple reflections between slopes) that can be formulated with a wide range of complexity. The diffuse component first includes the sky radiation, which has an angular distribution that depends on the atmospheric conditions (e.g. type of clouds,

aerosols; Olyphant, 1986). Nevertheless, an isotropic sky is often assumed even though simple equations with some angular dependence exist for single-layered atmospheres (Dozier, 1980). More advanced calculations can be performed numerically using multi-layer plane-parallel models, or Monte-Carlo models accounting for 3D cloud effects (Cornet et al., 2010). The greatest difficulty in practice is to obtain the atmospheric parameters to drive the models. The diffuse component also includes the illumination from surrounding mountains, which potentially results from multiple reflections between the surrounding and

the atmosphere or the surrounding and the considered point (Lenot et al., 2009). The reflections from surrounding mountains





can be treated assuming a simple reflection (e.g. assuming a constant slope, or purely diffuse radiation), up to full-featured Monte-Carlo models accounting for all possible ray trajectories (Lee et al., 2011). In the present paper, we aim to develop simple and computationally efficient correction methods, and to this end analytical formulations are preferred at the cost of simplifying assumptions. In particular, we neglect the multiple interactions with the atmosphere, but we do consider the

illumination from the neighbourhood surfaces, which is increasingly important as the slope increases.

The objective of the present paper is to 1) provide the theoretical framework to relate spectral apparent albedo to intrinsic albedo on a slope, and 2) present four correction methods to be applied depending on the available information and the assumptions that can be reasonably made, for different measurement conditions.

We first describe the theory of apparent albedo over a snow-covered slope (Section 2), then present the correction methods

(Section 3) and data used to evaluate the theory and the methods (Section 3.4). The theory is then applied to highlight the impact of the slope on apparent albedo spectra for various slope configurations (Section 4.1) and is compared to measured albedo from an Alpine site (Section 4.2). The correction for single acquisitions and timeseries are presented in Section 4.3. The Section 5 summarises and discusses the results.

The theoretical equations are implemented in an open-source computer code and a web application is made available to

interactively explore the slope effect in complement of the present paper (see Section 'code availability').

## 2   Theory

The objective is to relate the albedo measured with perfectly horizontal sensors over a slope to given intrinsic direct and diffuse albedo of the snow surface and other variables (diffuse direct proportion of the incident radiation, solar angles, slope angles, ...). To this end, we establish the equations for the downwelling and upwelling radiative fluxes on the slope and the neighbourhood

(Section 2.1) and on the upward and downward looking sensors (Section 2.2), to finally compute the apparent albedo as the ratio of the sensor measurements (Section 2.3). Further lengthy mathematical derivations can be found in the Appendix.

The geometry and angles of the problem are depicted in Fig. 1. The sensor height is considered to be very small with respect to the size of the slope and the horizontal surface. Both are indeed considered as semi-infinite planes for most equations, except when otherwise stated. The sensor is thus located very far from the horizontal surface. The theoretical equations are provided

for a given wavelength $\lambda$ and using trigonometric angles (azimuth is given anti-clockwise from the $x$ axis). The measurements and practical examples shown in Section 4 use geographic angles instead (sun azimuth and slope aspect are measured clockwise from the North).

### 2.1   Incoming and reflected radiation at the surfaces

The incoming light irradiance $E_\lambda^\downarrow$ on an horizontal surface or on a perfectly levelled upward-looking sensor is the sum of the

direct solar irradiance $E_\lambda^{\mathrm{sun}}$ weighted by the interception probability $\cos\theta_i$ ($\theta_i \in [0, \frac{\pi}{2}]$), and the diffuse irradiance coming from the sky $E_\lambda^{\mathrm{sky}}$:

$$E_\lambda^\downarrow(\theta_i, \phi_i) = E_\lambda^{\mathrm{sun}}(\theta_i, \phi_i)\cos\theta_i + E_\lambda^{\mathrm{sky}}. \tag{1}$$




where $\theta_i$, $\phi_i$ are the zenith and azimuth solar angles.

A slope is modelled here as a plane defined by its normal $N$ pointing upward with zenith angle $\theta_n$ and azimuth angle $\phi_n$. This surface receives direct solar radiation with a modified interception probability $\cos\acute{\theta}_i$, accounting for the local incidence angle, given by:

$$\cos\acute{\theta}_i = S\left(\cos\theta_i \cos\theta_n + \sin\theta_i \sin\theta_n \cos(\phi_i - \phi_n)\right) \tag{2}$$

where the "shadow" function $S(x)$ is 1 for $x > 0$ and 0 for $x \leq 0$. This ensures that when the sun is below the slope, the interception probability is null, and not negative.

In addition, the slope receives diffuse irradiance from the atmosphere $\acute{E}_\lambda^{\mathrm{sky}}$, which is lower than compared to the horizontal surface owing to the screening by the slope itself. Assuming isotropic sky radiation, the received diffuse irradiance can be written:

$$\acute{E}_\lambda^{\mathrm{sky}} = \iint\limits_{HS,\cos\acute{\theta}>0} \frac{E_\lambda^{\mathrm{sky}}}{\pi} \cos\acute{\theta} \sin\theta d\theta d\phi \tag{3}$$

where $\cos\acute{\theta}$ is given by Eq 2 without the $i$ subscript. The integral runs over the sky sector, i.e. the sector of the hemisphere (HS, $\theta \in [0, \frac{\pi}{2}]$ and $\phi \in [0, 2\pi]$) above the slope ($\cos\acute{\theta} > 0$, blue shape in Fig. 1 b). This integral can be calculated in the reference frame of the slope (Wang et al., 2016; Dumont et al., 2017) involving variable changes and some analytical calculations. An alternative is to write this integral in terms of the flux of the slope normal $N$ through the sky sector:

$$\acute{E}_\lambda^{\mathrm{sky}} = \frac{E_\lambda^{\mathrm{sky}}}{\pi} \iint\limits_{UHS,\cos\acute{\theta}>0} NdS. \tag{4}$$

Let us consider the closed surface including the sky sector (blue shade in Fig. 1 b), the unit half disk on the slope above the sensor and the unit half disk on the horizon plane. According to Gauss' theorem, the total flux of any conservative vector, as $N$, through this closed surface is null. Thus the flux in Eq. 4 can be deduced from the fluxes through both half disks. The flux through the half disk on the slope – which is perpendicular to $N$ is $-\frac{\pi}{2}$ and the flux through the half disk on the horizontal plane is $-\frac{\pi}{2}N.z = -\frac{\pi}{2}\cos\theta_n$, $z$ being the vertical axis. It follows that:

$$\acute{E}_\lambda^{\mathrm{sky}} = V E_\lambda^{\mathrm{sky}} \tag{5}$$

$$V = \frac{1 + \cos\theta_n}{2}. \tag{6}$$

$V$ can be interpreted as the cosine-weighted viewing fraction of the sky seen from the slope.

This reduced contribution from the atmosphere ($\acute{E}_\lambda^{\mathrm{sky}}$) is compensated by radiation coming from the solid angle under the horizon and above the slope, which impacts both the slope and the downward looking sensor (the latter is shown as green shade in Fig. 2). This contribution – that we qualify hereinafter by neighbourhood – writes:

$$\acute{E}_\lambda^{\mathrm{neigh}} = (1 - V) E_\lambda^{\mathrm{neigh}} \tag{7}$$





where we have assumed isotropic radiation and that the surfaces are infinite planes, so that the solid angle of the slope seen from the horizontal surface is the same as the solid angle of the horizontal surface seen from the tilted surface. This implies that the same cosine-weighted viewing fraction $1 - V$ applies to both surfaces as illustrated in Fig. 1 b. Summing the direct and the two diffuse contributions, the total incident irradiance on the slope is:

$$\acute{E}_\lambda^\downarrow(\theta_i, \phi_i) = E_\lambda^{\mathrm{sun}} \cos\acute{\theta}_i + V E_\lambda^{\mathrm{sky}} + (1 - V) E_\lambda^{\mathrm{neigh}} \tag{8}$$

and the flux reflected by the slope is:

$$\acute{E}_\lambda^\uparrow(\theta_i, \phi_i) = \bar{\alpha}_\lambda^{\mathrm{dir}}(\acute{\theta}_i) E_\lambda^{\mathrm{sun}} \cos\acute{\theta}_i + \bar{\alpha}_\lambda^{\mathrm{diff}} \left( V E_\lambda^{\mathrm{sky}} + (1 - V) E_\lambda^{\mathrm{neigh}} \right) \tag{9}$$

where we have distinguished the direct-hemispherical reflectance $\bar{\alpha}^{\mathrm{dir}}(\theta)$ (direct albedo) and the hemispherical-hemispherical reflectance $\bar{\alpha}^{\mathrm{diff}}$ (diffuse albedo).

To continue, the term $E_\lambda^{\mathrm{neigh}}$ must be specified. Many scenarios are imaginable depending on the terrain topography and cover (snow, vegetation, ...) present in the neighbourhood. For the sake of simplicity, here we consider an horizontal surface and distinguish two cases for the cover (Fig. 2): (case D) the horizontal surface is dark with reflectivity equal to zero ($E_\lambda^{\mathrm{neigh,D}} = 0$) and (case S) the horizontal surface is covered by snow having the same properties as on the slope, i.e. the same reflectivity. Case S is more complex than case D as it depends on the radiation reflected by the slope. The neighbourhood contribution by

the snow-covered horizontal surface can indeed be written as:

$$E_\lambda^{\mathrm{neigh,S}} = \bar{\alpha}_\lambda^{\mathrm{dir}}(\theta_i) E_\lambda^{\mathrm{sun}} \cos\theta_i + \bar{\alpha}_\lambda^{\mathrm{diff}} \left( V E_\lambda^{\mathrm{sky}} + (1 - V) \acute{E}_\lambda^\uparrow(\theta_i, \phi_i) \right). \tag{10}$$

This equation depends on the light flux reflected by the slope ($\acute{E}_\lambda^\uparrow(\theta_i, \phi_i)$), that itself depends on the neighbourhood irradiance (Eq 9). This interdependence is due to the mutual re-illumination by the two surfaces. We assume that radiation received from the slope is isotropic (as in Eq 3), meaning that snow is considered as a lambertian surface. This very common assumption

makes possible the analytical calculations presented in the following. However, we also assume that the direct albedo has an angular dependency ($\bar{\alpha}^{\mathrm{dir}}(\acute{\theta}_i)$). These two assumptions are physically incompatible, as the albedo of a strictly lambertian surface has no angular dependence. Nonetheless, the angular dependence of snow albedo has been evidenced by numerous ground observations (e.g. Dumont et al., 2017; Larue et al., submitted) and must be kept. On the other hand the lambertian assumption is required to conduct analytical calculations and provide simple formulations. This is the reason why in the

following we keep these two somewhat physically-incompatible assumptions.

Multiplying Eq 10 by the mutual re-illumination factor $M_\lambda$ defined by:

$$M_\lambda = (1 - V) \bar{\alpha}_\lambda^{\mathrm{diff}} \tag{11}$$

and adding the result to Eq. 9 give the upwelling light flux for the case S:

$$\acute{E}_\lambda^{\uparrow,S}(\theta_i, \phi_i) = \frac{\left( \bar{\alpha}_\lambda^{\mathrm{dir}}(\acute{\theta}_i) \cos\acute{\theta}_i + \bar{\alpha}_\lambda^{\mathrm{dir}}(\theta_i) M_\lambda \cos\theta_i \right) E_\lambda^{\mathrm{sun}} + \bar{\alpha}^{\mathrm{diff}}(\lambda) V (1 + M_\lambda) E_\lambda^{\mathrm{sky}}}{1 - M_\lambda^2}. \tag{12}$$





Similarly, by multiplying Eq. 9 by $M_\lambda$, the neighbourhood contribution follows:

$$E_\lambda^{\text{neigh,S}} = \frac{\left(M_\lambda \bar{\alpha}_\lambda^{\text{dir}}(\acute{\theta}_i)\cos\acute{\theta}_i + \bar{\alpha}_\lambda^{\text{dir}}(\theta_i)\cos\theta_i\right)E_\lambda^{\text{sun}} + \bar{\alpha}_\lambda^{\text{diff}}V(1+M_\lambda)E_\lambda^{\text{sky}}}{1 - M_\lambda^2} \tag{13}$$

which is symmetrical to Eq. 12 where the role of the slope and the horizontal surface are permuted.

## 2.2 Upwelling and downwelling radiation on the sensor

The flux received by the downward-looking horizontal sensor placed over a slope comes from the slope and the facing horizontal surface and follows:

$$I_\lambda^{\text{d}}(\theta_i, \phi_i) = V\acute{E}_\lambda^{\uparrow}(\theta_i, \phi_i) + (1-V)E_\lambda^{\text{neigh}}. \tag{14}$$

For the irradiance received by the upward-looking sensor, two cases shall be distinguished (Fig. 2). A first case is when the measurement is taken on the slope far from the horizontal neighbourhood and far from the top of the slope (hereinafter case mid-slope or M). In such a case, the part of the slope above the sensor reflects radiation toward the upward looking sensor (orange shade in Fig. 2) which adds up to the solar direct and sky radiation. The irradiance in the case M is written:

$$I_\lambda^{\text{u,caseM}}(\theta_i, \phi_i) = E_\lambda^{\text{sun}}S(\cos\acute{\theta}_i)\cos\theta_i + VE_\lambda^{\text{sky}} + (1-V)\acute{E}_\lambda^{\uparrow}(\theta_i, \phi_i) \tag{15}$$

where the shadow term $S(\cos\acute{\theta}_i)$ is 1 when the sun directly illuminates the slope and 0 otherwise.

The second case (called T) is when the sensor is above or close to the top of the slope, but still low enough for the downward looking sensor to mostly view the slope. This is a common case on slightly slopes. Rigorously, the notion of "top of the slope" is incompatible with the assumption of infinite slope used before, but it is acceptable here as trade-off between conducting analytical calculations and representing concrete practical situations. In case T, the irradiance received by the upward-looking sensor is given by:

$$I_\lambda^{\text{u,caseT}}(\theta_i, \phi_i) = E_\lambda^{\text{sun}}\cos\theta_i + E_\lambda^{\text{sky}}. \tag{16}$$

From this point, we have all the fluxes required to compute the albedo for 4 various cases (DT, DM, ST, and SM).

## 2.3 Apparent albedo

The measured, apparent, albedo can now be expressed as a function of the intrinsic direct and diffuse albedo, the diffuse-to-total ratio and the geometrical parameters. We present here the analytical derivation for the case of "small slopes" and leave the general case of "large slopes" in the Appendix. A summary of all the results is provided at the end of the section.

The small slope approximation mathematically corresponds to neglecting second order variations in $\theta_n$ (practical upper bounds for $\theta_n$ are given in Section 4), so that $\sin\theta_n \sim \theta_n$, $\cos\theta_n = 1 + o(\theta_n)$. It follows that $V = 1 + o(\theta_n)$ and that for all the cases considered above (cases S or D and T or M), the measured albedo reduces to the same mathematical form:

$$\acute{\alpha}_\lambda^{1^{\text{st}}}(\theta_i) = (1-r_\lambda)K(\theta_i, \acute{\theta}_i)\bar{\alpha}_\lambda^{\text{dir}}(\acute{\theta}_i) + r_\lambda\bar{\alpha}_\lambda^{\text{diff}}, \tag{17}$$



where we have introduced the geometrical factor

$$K(\theta_i, \acute{\theta}_i) = \frac{\cos \acute{\theta}_i}{\cos \theta_i}, \tag{18}$$

which is the main term carrying the first order slope effect, and the ratio $r_\lambda$ between incoming diffuse and total flux far above the slope:

$$r_\lambda = \frac{E_\lambda^{\mathrm{sky}}}{E_\lambda^{\mathrm{sun}} \cos \theta_i + E_\lambda^{\mathrm{sky}}}. \tag{19}$$

This ratio can be computed with an atmosphere radiative transfer model (e.g. SBDART, Ricchiazzi et al. (1998), 6S Vermote et al. (1997)) or can be measured with an upward-looking sensor by obstructing the direct sun light under the small slope approximation (as long as $V \approx 1$).

For large slopes, the mathematical derivation of the apparent albedo is more complex because $V$ becomes as important as $K$, i.e. the partial screening of the sky by the slope and the illumination by the neighbourhood have an increasing contribution. The mathematical details are given in the Appendix. The cases DT, DM, ST, and SM lead to different equations. Nevertheless, despite the relatively higher complexity compared to small slopes, a common general form can be found when the diffuse-to-total ratio $\acute{r}_\lambda$ is measured at the same location as the albedo. This form writes:

$$\acute{\alpha}_\lambda(\theta_i) = (1 - \acute{r}_\lambda) A_\lambda^{\mathrm{dir}}(\theta_i) + \acute{r}_\lambda A_\lambda^{\mathrm{diff}} \tag{20}$$

and it appears to be also suitable for flat surfaces and small slopes as well. Table 1 gives the $A^{\mathrm{dir}}$ and $A^{\mathrm{diff}}$ for all the cases considered in the present paper.

INSERT TABLE 1 HERE

## 3 Methods

We propose four methods to retrieve the surface albedo based on measured albedo on moderately slopes, when the small slope approximation applies (Section 2.3). We consider several cases depending whether the slope parameters are known or not, and in the latter, more complex case, depending on the additional available information or assumptions. In all cases, we assume that the diffuse-to-total ratio of incoming irradiance is known, which is a critical information to be able to perform the correction. Lastly, we assume that the albedo angular dependence is given by the Asymptotic Approximation Radiative Transfer (Kokhanovsky and Zege, 2004):

$$\bar{\alpha}_\lambda^{\mathrm{diff}} = \exp\left(-\sqrt{a_\lambda}\right) \tag{21}$$

$$\bar{\alpha}_\lambda^{\mathrm{dir}}(\theta) = \exp\left(-\frac{3}{7}(1 + 2\cos\theta)\sqrt{a_\lambda}\right) \tag{22}$$

where $a_\lambda$ is a factor depending on snow microstructure, ice absorption coefficient, impurity content and absorption, etc. Even if this factor is unknown, it is possible to relate the direct and diffuse albedos as follows:

$$\bar{\alpha}_\lambda^{\mathrm{dir}}(\theta) = \left[\bar{\alpha}_\lambda^{\mathrm{diff}}\right]^{n(\theta)} \text{ with } n(\theta) = \frac{3}{7}(1 + 2\cos\theta). \tag{23}$$





**Table 1.** Apparent albedo formulation $\acute{\alpha}(\theta_i) = (1 - \acute{r})A^{\mathrm{dir}}(\theta_i) + \acute{r}A^{\mathrm{diff}}$ for horizontal terrain, small slopes and for 4 configurations with significant slopes depending on whether the neighbourhood is covered by snow or by a dark surface, and the albedo and diffuse-to-total ratio measurements are taken mid-slope or at the top of the slope). The slope is not in its own shadow $S(\cos\acute{\theta}_i) > 0$. We have also introduced $M = (1 - V)\bar{\alpha}^{\mathrm{diff}}$. The $\lambda$ dependence is implicit for sake of simplicity.

| Case | $A^{\mathrm{dir}}(\theta_i)$ | $A^{\mathrm{diff}}$ |
|---|---|---|
| no slope | $\bar{\alpha}^{\mathrm{dir}}(\theta_i)$ | $\bar{\alpha}^{\mathrm{diff}}$ |
| small slope | $K\bar{\alpha}^{\mathrm{dir}}(\acute{\theta}_i)$ | $\bar{\alpha}^{\mathrm{diff}}$ |
| dark & top (DT) | $VK\bar{\alpha}^{\mathrm{dir}}(\acute{\theta}_i)$ | $V^2\bar{\alpha}^{\mathrm{diff}}$ |
| dark & mid-slope (DM) | $\frac{V}{1+M}K\bar{\alpha}^{\mathrm{dir}}(\acute{\theta}_i)$ | $\frac{V}{1+M}\bar{\alpha}^{\mathrm{diff}}$ |
| snow & top (ST) | $\frac{V + M(1-V)}{1 - M^2}K\bar{\alpha}^{\mathrm{dir}}(\acute{\theta}_i) + \frac{MV + (1-V)}{1 - M^2}\bar{\alpha}^{\mathrm{dir}}(\theta_i)$ | $\frac{V}{1 - M}\bar{\alpha}^{\mathrm{diff}}$ |
| snow & mid-slope (SM) | $\frac{V}{1 + M}K\bar{\alpha}^{\mathrm{dir}}(\acute{\theta}_i) + \frac{1 - V + M}{1 + M}\bar{\alpha}^{\mathrm{dir}}(\theta_i)$ | $\bar{\alpha}^{\mathrm{diff}}$ |

The methods presented in the following solely depend on this relationship between the direct and diffuse albedos, they do not explicitly depend on $a_\lambda$ and all the hidden complexity in it. Furthermore they can be easily adapted to other formulations for $n(\theta)$.

### 3.1 Albedo correction with known slope parameters

5 Given the slope parameters ($\theta_n$ and $\phi_n$), measured $\acute{\alpha}_\lambda^{\mathrm{mes}}$, and measured $r_\lambda$, the goal is to estimate $\bar{\alpha}_\lambda^{\mathrm{diff}}$, using the relationship between the direct and diffuse albedos and the small slope formulation (Eq. 17) given the following equation:

$$\acute{\alpha}_\lambda^{\mathrm{mes}} = (1 - r_\lambda)K(\theta_i, \acute{\theta}_i)\left[\bar{\alpha}_\lambda^{\mathrm{diff}}\right]^{n(\acute{\theta}_i)} + r_\lambda\bar{\alpha}_\lambda^{\mathrm{diff}} \tag{24}$$

where the only unknown is $\bar{\alpha}_\lambda^{\mathrm{diff}}$. It can be solved with any non-linear equation solver. For instance, rearranging this equation to let the difference between the direct and diffuse albedos appear, leads to a solution that can be efficiently solved by iterations

10 as follows:

$$\bar{\alpha}_\lambda^{\mathrm{diff}}(j+1) = \frac{\acute{\alpha}_\lambda^{\mathrm{mes}} - (1 - r_\lambda)K\left(\bar{\alpha}_\lambda^{\mathrm{diff}}(j) - \left[\bar{\alpha}_\lambda^{\mathrm{diff}}(j)\right]^{n(\acute{\theta}_i)}\right)}{(1 - r_\lambda)K + r_\lambda} \tag{25}$$

with $j$ the iteration counter, starting at 0 with $\bar{\alpha}_\lambda^{\mathrm{diff}}(j=0) = \min(\acute{\alpha}_\lambda^{\mathrm{mes}}, 1)$. The convergence has been tested for a wide range of parameters. Even in the worst case (i.e. the slope opposed to the sun, grazing zenith incidence angle, $K = 0.2$), 10 iterations are sufficient to reach a precision of 0.1%. In most practical cases ($K$ close to 1), less than 5 iterations are sufficient.



## 3.2 Albedo correction with unknown slope parameters

The slope parameters are often unavailable or the precision on these parameters is insufficient. If we further assume that surface snow contains negligible amounts of light absorbing impurities, the intrinsic albedo in the visible wavelengths is nearly constant and close to a value of 1 over a wide range, typically between $400-500\,\mathrm{nm}$ (Warren and Wiscombe, 1980). This range is also
where the slope effect on albedo is the most visible – which is highlighted in Section 4 – so that constraining $\bar{\alpha}_\lambda^{\mathrm{diff}} = \alpha_0$ for a range of wavelengths provides a way to estimate $K$. Indeed, Eq. 24 becomes linear in $K$ if $n$ is calculated by approximating $\acute{\theta}_i$ by $\theta_i$, an approximation that is more and more valid as $\alpha_0$ gets closer to 1. One albedo measurement at one wavelength is in principle sufficient to estimate $K$, but to improve the reliability, we consider here multiple measurements at $L$ different wavelengths $\lambda_l$ in the range $400-500\,\mathrm{nm}$. The least-square optimal solution of the linear equation with the unknown $K$ gives
the estimate $\tilde{K}$ according to:

$$\tilde{K} = \frac{\sum_{l=1}^{N_\lambda} (\acute{\alpha}_{\lambda_l}^{\mathrm{mes}} - r_{\lambda_l}\alpha_0)(1-r_{\lambda_l})}{\sum_{l=1}^{N_\lambda}(1-r_{\lambda_l})^2 \alpha_0^{n(\theta_i)}}. \tag{26}$$

Here we use $\alpha_0 = 0.98$. From $\tilde{K}$, it is possible to estimate $\tilde{n}(\acute{\theta}_i) = \frac{3}{7}(1 + 2\tilde{K}\cos\theta_i)$ which is sufficient to apply the iterative method depicted in Eq. 25 and hence to obtain the diffuse albedo. Note however that knowing $K$ is insufficient to estimate the slope angles $\theta_n$ and $\phi_n$.

## 3.3 Unconstrained and constrained correction of diurnal cycle of albedo with unknown slope parameters

Another practical case is when albedo is measured at different hours during a single day. If we can assume that snow properties have not evolved during that day (e.g. no precipitation, no melt), $\bar{\alpha}_\lambda^{\mathrm{diff}}$ is a constant. This assumption leads to the following system of equations:

$$\acute{\alpha}_\lambda^{\mathrm{mes}}(t) = (1 - r_\lambda(t))\frac{\cos\acute{\theta}_i(t)}{\cos\theta_i(t)} \left[\bar{\alpha}_\lambda^{\mathrm{diff}}\right]^{\frac{3}{7}(1+2\cos\theta_i(t))} + r_\lambda(t)\bar{\alpha}_\lambda^{\mathrm{diff}} \tag{27}$$

$$\cos\acute{\theta}_i(t) = S[\cos\theta_i(t)\cos\theta_n + \sin\theta_i(t)\sin\theta_n\cos(\phi_i(t) - \phi_n)]. \tag{28}$$

where $\bar{\alpha}_\lambda^{\mathrm{diff}}$ is the unknown along with the two slope parameters $\theta_n$ and $\phi_n$. Assuming that albedo measurements are taken at $N_\lambda$ wavelengths, and $N_t$ time steps, the number of equations is $N_\lambda N_t$ and the number of unknowns is $N_\lambda + 2$. Because of the strong non-linear coupling between the unknowns, no simple analytical method is devisable, but the system can be numerically solved with non-linear least-squares using the cost function:

$$\mathcal{J} = \frac{1}{2}\sum_{l=1}^{N_\lambda}\sum_{j=1}^{N_t}\left(\acute{\alpha}_{\lambda_l}^{\mathrm{mes}}(t_j) - \acute{\alpha}_{\lambda_l}^{\mathrm{mod}}(t_j)\right)^2 \tag{29}$$

where $\acute{\alpha}_{\lambda_l}^{\mathrm{mod}}(t_i)$ is given by the right term in Eq. 27. Minimising the cost function in this study is performed using the Python function scipy.optimize.leastsq implementing the Levenberg-Marquardt algorithm (Levenberg, 1944).





Another method is derived for situations where the snow surface is known to be free of impurities. In such a case, it may be interesting to constrain the albedo value in the blue-green range, $\bar{\alpha}_\lambda^{\mathrm{diff}} = \alpha_0, \lambda \in [400\,\mathrm{nm}, 500\,\mathrm{nm}]$, as in Section 3.2. Adding such a constraint is straightforward using the Python function scipy.optimize.minimize.

## 3.4 Data

Albedo data were acquired at the Col du Lautaret site (45°2'4"N, 6°24'18"E) in the framework of the EBONI campaign running from 2016 to 2019, already presented in Tuzet et al. (2019) and Larue et al. (submitted). The albedo data were collected at different locations with various slope configurations in an overall snowy environment, using the manually-operated albedometer Solab, and at a fixed position on a south-east facing slope using the automatic spectrometer Autosolexs.

Solab is composed of a single light collector fixed at the tip of a 3-m long arm, and connected to a spectrometer (400–
1050 nm) (Belke-Brea et al., 2019; Tuzet, to be submitted). Downwelling and upwelling fluxes required to compute the albedo are successively acquired by manually rotating the arm held horizontal, hence pointing the collector upward, then downward. The albedo acquisition is only considered valid if the solar variations stay within 0.1% during this operation, which takes no more than 30 s. Levelling of the collector, which is critical for the quality of the data, is adjusted and maintained by the operator during the measurements using an electronic inclinometer fitted adjacent to the light collector. The accuracy and stability are
usually better than 0.2°. Sensor height is about 1 m. The processing of the raw spectra to compute albedo is detailed in Picard et al. (2016b). Repeatability of the measurements in clear-sky conditions is better than 1%. In addition, the diffuse-to-total ratio is measured by first recording the total downwelling flux, as for the albedo measurement, and second the diffuse downwelling flux by shading the collector from the direct sun using a thin black metallic strip fixed to the arm. The ratio is calculated following the same processing steps as for the albedo. Terrain slope is measured after the Solalb acquisition using a 3-m long
and 5-cm wide bar fitted with an electronic inclinometer. In 2018 when Solalb data were obtained, the greatest slope was sought by rotating the bar on the surface until the maximum inclination was found. The azimuth of the bar, giving the aspect of the slope, was measured with a handheld compass. In 2019, at the Autosolexs location, the slope was measured at the end of the season only, by taking two inclination measurements with the bar lying in the north-south and east-west directions respectively. The greatest slope and azimuth were deduced by calculation. Despite the inclinometer intrinsic accuracy (0.1°), the precision
of the slope and aspect angle obtained with these protocols is relatively mediocre (probably >1°) compared to the requirements for the albedo interpretation. It is also worth pointing out that the natural surface is not always a perfect plane, even at the scale of the bar (3 m), but the roughness was not recorded.

Autosolexs has two fixed light collectors pointing upward and downward, which are successively connected to a spectrometer by an optical switch (Picard et al., 2016b) every 12 min. The collectors and the spectrometer have the same specifications as
those in Solalb. Sensor height is about 2 m. Data processing, detailed in Picard et al. (2016b), also follows the same steps as for Solalb acquisitions, except that an additional cross-calibration step is required to account for the slightly different responses of the two collectors. In addition to albedo, Autosolexs automatically records the diffuse irradiance using a third light collector shaded by a small rod following the course of the sun. Data used here were acquired on 23 March 2018 in clear-sky conditions.





## 4 Results

### 4.1 Theoretical analysis of the apparent albedo formulations

#### 4.1.1 Quantitative impact of slope on apparent albedo

The impact of the slope is studied considering both 100% diffuse radiation (overcast conditions) and 100% direct radiation

for various zenith solar angles $\theta_i$. In all cases, the sun is located to the south. For an intrinsic direct albedo of 0.8, Fig. 3 shows the impact of the slope from -35°(north-facing) to 35°(south-facing). With direct illumination, the impact of the slope is considerable, and greatly increases with the solar zenith angle. For instance a sun-facing slope of 10° affects the albedo by +0.04, +0.13, and +0.42, at SZAs of 20°, 45° and 75° respectively. Neglecting the slope effect leads to detectable albedo errors of 0.01 or larger for slope inclinations larger than 2°, 0.7° and 0.3° at SZAs of 20°, 45° and 75° respectively. Such

slopes are very small and barely visible to the eye in the field, yet have a detectable effect on the albedo. At SZA=70°, which is typical of winter at mid-latitudes, the apparent albedo ranges from 0 to 1.6 over the investigated slope range, a two-fold variation with respect to the flat albedo value. At SZA=45°, which is a typical angle during the melt period in many regions, the apparent albedo ranges between 0.25 to 1.1 for the various formulations. Even when the sun is high (SZA=20°), albedo varies significantly from 0.4 to 0.9. Apparent albedo higher than 1.0 for the south-facing slopes may be surprising at a first

glance, but it is mainly the consequence of the $K$ factor which accounts for the higher interception probability of the sun beam by these slopes facing the sun compared to horizontal surfaces. It is clear that apparent albedo (being higher or even lower than 1) must not be used for energy budget calculations as it is commonly done for a flat terrain.

The differences between the scenarios appear on the graph starting from about a 10° slope (both north- and south-facing). For a slope of 35°, the maximum difference is 0.2 for SZA=45°and 0.4 for SZA=70°, which is very large. The lowest apparent

albedo in south-facing slopes is for surrounding dark surfaces and mid-slope measurements, because the downward looking sensor has a deficit of incoming radiation due to the dark surface, while the upward looking sensor has an excess due to downward reflections from the upper slope. The opposite is observed for measurements with surrounding snow taken near the top of the slope, because the downward looking sensor receives additional illumination from the horizontal area while the upward looking sensor is not affected by the slope.

The slope effect is found to be maximum under direct illumination which applies well to the near-infrared domain under clear-sky conditions. Under diffuse radiation (overcast conditions or in the blue and U.V.) the slope effect is null for the small slope formulation, very weak when the surrounding surfaces are covered by snow, and leads to a decreased albedo for a dark neighbourhood, but this decrease is generally smaller than the impact under direct illumination.

The dependence of the albedo variations with slope on the type of illumination has a practical interest to understand spectral

albedo measured in natural conditions, which is addressed in the next section.



### 4.1.2 Spectral shape of the apparent albedo on a slope

Under clear-sky or partially cloudy conditions, the proportion of direct and diffuse incident radiation varies as a function of the wavelength, and given the contrasted response to the slope between direct and diffuse albedo, the shape of measured albedo spectra is distorted over slope. Figure 4 illustrates this distortion by showing theoretical apparent albedo spectra for various

slopes under clear sky-conditions. The calculation here assumes pure Rayleigh diffusion, with a given diffuse-to-total ratio modelled as $r_\lambda = (\lambda_0/\lambda)^n$ with $n = 4$ and $\lambda_0 = 350\,\mathrm{nm}$. The snowpack is considered to be homogeneous and infinitely deep. The specific surface area (SSA) is set to $20\,\mathrm{m}^2\,\mathrm{kg}^{-1}$, a typical value for superficial snow in Alpine regions in winter (Dumont et al., 2017) and on the Antarctic Plateau in late Summer (Libois et al., 2015). The snow is free of any impurities. To relate the direct and diffuse albedo to snow physical properties, we rely on the Asymptotic Radiative Transfer theory (Eq. 21) where:

$$a_\lambda = 16\frac{4\pi n_\lambda''}{\lambda}\frac{B}{3(1-g)}\frac{1}{\rho_{\mathrm{ice}}\mathrm{SSA}} \tag{30}$$

with the ice permittivity imaginary part $n_\lambda''$ taken from Picard et al. (2016a), the ice density $\rho_{\mathrm{ice}} = 917\,\mathrm{kg}\,\mathrm{m}^{-3}$ and the shape factors $B = 1.6$ and $g = 0.845$ according to Libois et al. (2014).

The flat surface case (second panel in Fig. 4) shows a typical albedo spectrum (Warren and Wiscombe, 1980) with a nearly constant value in the blue and green (400–550 nm), and a decreasing trend at longer wavelengths attributed to the increasing

ice absorption (Warren and Brandt, 2008; Picard et al., 2016a). The absorption feature at 1030 nm is more marked than those at shorter wavelengths (800 and 890 nm) and leads to a local minimum which is often used to infer grain size metrics (Painter et al., 2007; Kokhanovsky et al., 2019). Because we have chosen $\theta_i$=45°, the direct and diffuse components (dash gray curves) are similar, and as a consequence, are overlaid by the apparent albedo spectrum (blue curve) that is a weighted proportion of both components. For other $\theta_i$ angles, despite a potentially larger difference between the diffuse and direct albedo, one would

observe the same plateau in the blue and green with a value close to 1 for flat terrain.

When the terrain is not flat, the shape of the albedo spectrum is affected, and not only depends on the ice absorption variations but also on the proportion of direct and diffuse illumination. In the red and infra-red (>700 nm), because the illumination is mostly direct, we observe that the apparent albedo is close to the direct albedo, and is therefore very sensitive to the slope as shown in the previous section. It is lower for a north-facing slope compared to the albedo over a flat terrain and is larger

for a south-facing slope. In the blue (close to 400 nm) where scattering by the atmosphere is significant, the apparent albedo spectrum lies between the direct and the diffuse albedo curves, and because the diffuse albedo is always close to 1 for small slopes, the apparent albedo also tends to 1. In the intermediate range, between the blue and the red, the variations of albedo with wavelength are driven by the transition from diffuse-dominant to mostly-direct illumination rather than by the variations of the ice absorption. The result is a distortion of the spectra shape that depends on the slope and on the illumination conditions.

The distortion is very different for north- and south-facing slopes. For a north-facing slope, where the direct albedo is reduced by the slope and is thus systematically lower than the diffuse albedo, the apparent albedo shows a marked decrease from the blue to the red (left column in Fig. 4. Conversely for a south-facing slope, the albedo first increases with wavelength due to the increasing proportion of direct illumination and then decreases because the ice absorption increases rapidly enough to





dominate the trend (right column and bottom middle in Fig. 4). The albedo for a south-facing slopes thus presents a maximum value in the transition wavelength range, which is always above 1, around 1.008 at 490 nm for a 2° slope, 1.09 at 570 nm and 1.18 at 600 nm (values taken from the small slope calculation). These values depend on the illumination conditions and snow characteristics, as lower $n$, $\lambda_0$, $\theta_i$ or SSA lead to a lower maximum. The strongest distortion of the spectra is observed for steep

slopes, low sun elevation, low aerosol content and high altitude. It is noteworthy that all the approximations accounting for the slope effect yield the same general distortion, only small differences appear from about 10° and becomes significant from about 20°, as shown in the previous Section and in Fig. 3.

## 4.2   Comparison between theoretical calculations and albedo measurements

A set of Solalb measurements acquired in Winter 2018 on clean and dry snow has been selected to cover a wide range of

slope inclinations and orientations. The comparison with the theoretical apparent albedo using measured slopes, SSA and diffuse-to-total ratio is shown in Fig. 5 (left column, sorted by increasing slope) for the small slope approximation and the SM case. The latter was chosen among the four large-slope cases because the surroundings were fully covered by snow and the measurements were taken mid-slope. The agreement is variable but in general, the measured (apparent) albedo spectra are clearly affected by the slope as predicted by the theory. All the cases show that the measured albedo is either lower or higher

than 1 accordingly to the slope aspect relative to the sun azimuth, that is, for cases 1, 2, 3, 5 and 7 that are opposite to the sun, the albedo is lower than 1 and features a decreasing trend in the blue–yellow, while for cases 4 and 6 facing the sun, the albedo is higher than 1 and features a maximum in the green–yellow. The root mean square error (RMSE) is in the range 0.015 – 0.04, the worst cases being the ones with the largest slope. The bias (simulated minus measured albedo) is variable between about -0.02 and 0.02 except for the largest slope case (case 7) where it reaches 0.04. At last we note that the difference

between the two approximations (small slope and SM case) is undetectable in all but case 7 with an 18° slope, and even in this case, the difference is much smaller than the discrepancies between the measured and calculated albedo. This indicates that the uncertainties in the albedo measurements and the ancillary data required to perform the calculation (slope, SSA, and diffuse-to-total ratio) are larger than the theoretical formulation differences, and therefore that improving the measurement accuracy is a higher priority than developing more advanced theories. As a corollary, using the simple small slope approximation is

probably sufficient for most applications.

Figure 6 shows a similar calculation for the Autosolexs measurements taken on 23 March 2018, a day with continuous clear-sky conditions. Since not all the measured and calculated spectra can be shown (acquisition every 12 min), we selected spectra at a few representative hours (panel a) and albedo time series at a few representative wavelengths (panel b). The slope estimated from a 50 cm resolution digital elevation model (DEM) of the Lautaret area gives $\theta_n = 4.5°$ and $\phi_n = 165°$.

However, the calculation using these angles disagrees with the observations, so that we increased the slope inclination up to $\theta_n = 7.5°$ to obtain the agreement shown in Fig. 6. A possible explanation is the difference between the snow surface and the DEM that was acquired over bare ground, but we did not perform in-situ slope and aspect measurements in the footprint of the sensor, to avoid snow disturbance. This issue emphasises the difficulty to acquire in-situ slope measurements, which are valuable to perform albedo corrections or interpretations. In addition we choose SSA=30 $\mathrm{m^2\ kg^{-1}}$ to obtain an agreement





in the near-infrared (1030 nm). This value falls within the range of SSA measured a few days earlier in the surrounding (14 March, Fig. 5).

The slope being south-south-east, most albedo spectra are affected by sun-facing distortions, i.e. a maximum higher than 1 in the visible and a concave spectral shape, which is also predicted by the calculation. A greater distortion is observed in the morning when the sun is facing the slope, while the spectrum taken just before the local sunset is almost unaffected. The agreement between the measurements and the theory is overall good even though the adjustment of the slope inclination has a great influence on such a result. The largest discrepancy between measurements and calculations is found at the end of the day, in the near-infrared, suggesting that snow metamorphism during the day might have decreased the specific surface area. Additional simulations with a SSA of $22\,\mathrm{m^2\,kg^{-1}}$ (dotted line) solve this discrepancy at the end of the day.

## 4.3 Albedo correction

The correction methods described in Section 3 are applied to the measurements used in the previous section, first using the individual measurements taken with Solalb, and second the time-series of measurements at a single location taken with Autosolexs.

### 4.3.1 Albedo correction with known and unknown slope parameters

Figure 5 (second and third columns) shows corrected albedo using the measured diffuse-to-total ratio, assuming a particular angular dependence (Eq. 23) and using the measured slope inclination and aspect (in the second column) or considering unknown slope parameters (in the third column). The quality of the correction is judged according to 1) the resemblance of the shape of the corrected albedo spectrum to a flat-terrain albedo and in particular whether the albedo is constant and close to 1 in the blue-green range and 2) on the adequacy of the theoretical flat-terrain albedo calculated using measured SSA to fit the corrected albedo in the near infra-red domain (distance of the violet or red curve from the gray curve). In all the cases, the corrected albedo is closer to a flat-terrain albedo spectrum than the measured one, meaning that the correction improves the measurements. In general, the correction without using the slope parameters yields nearly flat albedo with value lower than 1 in the blue-green, meaning that even a highly distorted albedo can be recovered. The correction method also achieves better results than when using the actual measured parameters (all except case 2, and case 5 which shows a marginal deterioration) highlighting the difficulty to acquire sufficiently accurate slope parameters. However, the slope uncertainty is not the only cause, for instance the shape of the measured spectra in case 2 is suspect in the range $400 - 600\,\mathrm{nm}$, suggesting a measurement artefact. Neither the forward simulation nor the corrections reproduce a "normal" spectral shape.

Overall, we also note that the quality of the correction does not depend on the slope inclination. The amplitude of the correction (i.e. the difference between the blue and the violet or red curves) can be large (e.g. case 7), still the correction is satisfactory. These results show that the correction of apparent albedo is possible for a relatively large range of slopes, even without knowing the slope parameters. However, this high performance relies on the assumption of clean snow and on the quality of the cross-calibration between the upwelling and downwelling flux acquisitions, which is usually not an issue for manual measurements (e.g. Carmagnola et al., 2013), because the same channel is used for both acquisitions, but could be for





dual-channel automatic spectrometers (e.g. Picard et al., 2016b). The method interpreting any departure from $\alpha_0 = 0.98$ in the blue–green as the presence of slope, the correction would be incorrect if this departure was to be caused by impurities, a poor cross-calibration or any other artefacts.

### 4.3.2 Albedo correction for diurnal cycle measurements with unknown slope parameters

Figure 7 shows Autosolexs albedo spectra corrected using the diurnal time evolution on 23 March 2018 with the unconstrained (blue) and constrained (orange) methods described in Section 3.3. All the spectra measured between 8:30 and 16:00 UTC (gray curves) are used as inputs of the methods. Since no in-situ SSA were measured near the albedometer it is not possible to compute the expected diffuse albedo that the correction methods are expected to approach. However, fully overcast conditions occurred 4 days later and assuming that the superficial snowpack did not change during this lapse of time, the albedo measured

then (green curves) provides a reference diffuse albedo to assess the performance of the correction.

The albedo spectrum corrected without constraint (blue curve) shows an improvement compared to the measured albedo spectra (gray), the albedo value is more constant in the visible and closer to 1 in the blue-green. Nevertheless, it is still higher than 1 and slightly concave, which is a clear sign of insufficient correction. Furthermore, the method produces variable results depending on the conditions (e.g. taking a subset of the inputs, results not shown). This unstable behaviour comes from the

fact that the only constraint to infer the slope parameters is provided by the temporal variations induced by the course of the sun. In turn, any small uncertainty that affects the angular behaviour of the sensor (e.g. imperfect levelling or imperfect cosine response), of the snow surface albedo (e.g. roughness) or of the underlying theory (mainly Eq. 23) may have significant consequences. In our case, the method appears unsuitable for radiative forcing calculations, which require at least albedo lower than 1, but it could be helpful when only wavelength band ratios are necessary (Mary et al., 2013). The case of small slopes

may also be more favourable, and remains to be explored.

Assuming an impurity-free surface brings a strong constraint on the albedo in the visible, and if it is true, precisely provides what was missing in the unconstrained method. The constrained method performs well in our case (orange versus green curves, in Fig. 7), especially in the visible. The slightly higher value in the near-infrared is not necessarily an error, as it could reflect a decrease of SSA during the 4 days between the clear-sky measurements (23 March) and overcast measurements (27 March).

Although the correction method only constrains the albedo value in the visible, the shape of the spectrum is almost perfectly corrected over the whole spectral range, and the spectrum is smooth (which could have been added as a constraint, but was not). The smoothing comes from the fact that all the measured spectra are effectively used in the estimation process, canceling the incoherent noise visible in the measurements (gray curves).

Along with the corrected albedo, both methods provide estimates of the two slope angles, which was not possible with

the methods using a single acquisition. We obtain a slope of 3° with the unconstrained method. This low value explains why the corrected albedo is still higher than 1 at some wavelengths and still has the traits of a sun-facing slope. In contrast, the constrained method estimates a slope of 7.6° and an aspect of 157° and achieves a good correction. Only the aspect agrees with the DEM (bare ground) of the area.





## 5 Discussion

### 5.1 Apparent albedo and energy conservation

The theoretical developments and the measured spectra presented throughout this paper make it clear that the "apparent albedo" of a slope is not an albedo *stricto senso*, and must not be used to compute the energy absorbed by the surface with the usual

equation $(1-\alpha)E^{\downarrow}$. This is obvious when the apparent albedo value is higher than 1, because the calculated absorption would then become negative, but this statement holds true even for values lower than 1. The reason is that the apparent albedo is not a pure reflective property of the surface, it also contains the factor converting the energy received on a flat surface to that received by the slope ($K$), as well as other terms in the case of large slopes. Only the reflective property is subject to energy conservation, $K$ is not. A correct way to calculate the absorption $A_\lambda$ per unit of flat ground surface area is:

$$A_\lambda = \frac{1}{\cos\theta_n}\left((1-r_\lambda)(1-\bar{\alpha}_\lambda^{\mathrm{dir}}(\acute{\theta}_i))K(\theta_i,\acute{\theta}_i)\cos\theta_i E_\lambda^{\mathrm{sun}} + r_\lambda(1-\bar{\alpha}_\lambda^{\mathrm{diff}})E_\lambda^{\mathrm{sky}}\right)$$
(31)

which requires more information than just the apparent albedo and the irradiance. It is thereby inapplicable in many situations such as when only the spectral albedo is measured. This equation respects the energy conservation principle even if $K > 1$, because the energy absorbed on the slope is always lower than the irradiance ($E_\lambda^{\mathrm{sun}} + E_\lambda^{\mathrm{sky}}$), owing to $K\cos\theta_i < 1$.

It is worth recalling that measured broadband albedo is similarly subject to the slope effect and that the usual equation

$(1-\alpha)SW^{\downarrow}$ is also invalid on a slope when $\alpha$ is measured with horizontal sensors, even though the measured value is lower than 1. Furthermore, the slope effect is impossible to detect from a single acquisition of broadband albedo.

Ideally, the term 'albedo' should not be used to refer to the ratio of upwelling and downwelling light fluxes over a slope, but in practice, we recommend to at least evoke this issue, systematically using the terms "apparent" or "measured".

### 5.2 Albedo accuracy and slope effect

All the results presented in Section 4 highlight the large and complex sensitivity of the apparent albedo to slope. If one targets an albedo measurement accuracy of 0.01 (Picard et al., 2016b) – for instance to detect small amount of impurities (Warren, 2013), to detect multi-annual trends (Dumont et al., 2014), or to close the surface radiative budget within 4-8 $\mathrm{Wm}^{-2}$ – the error on the slope inclination should remain below 0.6° (considering the sun at 45° zenith angle and not including aspect error, a very favorable case). This underlines the high level of measurement quality requirements.

Improvement of instruments (equipped with an inclinometer) and protocols (systematic acquisition of diffuse-to-total ratio and slope inclination) is a response to these requirements. Another response is the *ad hoc* correction of the measured albedo. In a first attempt, it was common in our community to normalise the measured albedo to 1 (or 0.98) in the visible (e.g. Picard et al., 2016b) because it helped overcome not only the slope effect but also other common problems, such as the cross calibration of the upwelling and downwelling channels, the poor cosine response of the sensors (Zibordi and Bulgarelli, 2007; Picard

et al., 2016b), or eliminate operator shadows (Carmagnola et al., 2013). However, this is an error-prone practice, because the spectrum of the scaled albedo on a slope looks similar to the albedo of dirty snow surface in the case of sun-facing slopes, as illustrated in Fig 8. The convex shape in the visible domain is indeed a common feature of both flat dirty snow and pristine





snow slopes. For instance a pristine snow surface with a small slope of $2°$ has a similar albedo to flat dirty snow surface with $61\,\mathrm{ng\,g^{-1}}$ of black carbon, if the apparent albedo measured on the slope is scaled down by 3% only (blue curve in Fig. 8), so that its maximum is equal to 0.98.

We thereby recommend not to apply rough scaling, but instead to use a proper correction method as described in this paper.
Nevertheless, the risk of over- or under-corrections also exists, and may result in the confusion with dirty snow. For instance over-correcting measurements taken over pristine snow on a slope facing away from the sun, results in a spectrum with a convex shape characteristic of dirty snow. Our conclusion is that small slopes, impurity content and calibration errors, sensor angular response and operator shadows are inter-related, and only a global assessment of the main error sources, and a consistent treatment of these sources can lead to properly corrected albedo spectra.

**5.3 The correction methods and their underlying assumptions.**

The results show that the correction of the slope effect is possible in the domain of validity of small slopes ($\lesssim 15°$) but the method to be applied depends on the available information about the actual illumination and slope, and/or some assumptions on the snow impurity content. Throughout the paper, we have assumed a known diffuse-to-total ratio, either because it was measured at the same level as the albedo measurements or as a fallback because it was calculated above the mountain ridge
with an atmospheric radiative transfer model. Knowing this ratio is absolutely required to correct albedo in the visible domain, except in the very particular case of a solar zenith angle close to the effective angle of the diffuse illumination (in the AART formulation, this occurs when $n(\theta) \approx 1$ and is usually $45$–$50°$). In the near-infrared domain under clear sky conditions, the diffuse illumination can be neglected and if the variation of SSA can also be neglected, a correction without diffuse-to-total ratio is feasible.

Here we used systematically diffuse-to-total measurements acquired at almost the same time as the albedo acquisitions, which is a very favourable case. Weiser et al. (2016) use nearby measurements of the ratio, but considered a constant value for all clear-sky days. The solution to use an atmospheric model is attractive when no measurements are available. Tuzet (to be submitted) obtained consistent results using the SBDART model (Ricchiazzi et al., 1998) parameterised with a generic mid-latitude winter atmosphere and only considering solar zenith angle variations, under the condition that clear-sky conditions are
separated from overcast conditions. Dumont et al. (2017) applied the same model but used actual atmospheric profiles obtained from a meteorological analysis and cloud optical depth adjusted using shortwave in-situ measurements. The trade-off between costs and benefits of using such a complex setting, and more generally the impact of the accuracy of the diffuse-to-total ratio remain to be assessed.

When a single albedo acquisition is available, the most favorable case is when the slope angles (inclination and aspect) or
the slope parameter ($K$ or the local solar zenith angle) are precisely known. The correction then requires no assumption on the snow surface (works for clean snow and snow covers with any impurity content). The main required assumption is the angular dependence of the direct albedo (Eq. 23) which depends on the theory used (here AART), and on the smoothness of the surface. However, our results show that the quality of the correction is variable and we suspect the slope parameters' accuracy to be an issue (Fig. 5, second column). Assuming that snow is clean strongly constrains the correction and relaxes the requirement





of knowing the slope parameters (Fig. 5, third columns). When the SSA is known with precision, it should also be possible to devise a correction method without the slope parameters (not covered in this paper). At last, the case of dirty snow, unknown SSA and unknown slope parameters cannot be solved with a single acquisition.

When multiple albedo acquisitions at the same location are available for a wide range of sun positions (e.g. time series of

albedo), a correction method exists assuming constant snow properties throughout the acquisitions but without the requirement of assuming clean snow or a known SSA (Eq. 29). The variations of the distortion of the spectrum carry the signature of the slope effect, enabling in principle to recover both slope inclination and aspect. The method should best work when the sun azimuth covers a wide range such as in the polar regions during the summer. In mid-latitude regions the range is more limited (less than 180°) which may explain the mediocre results we obtained with this method (Fig. 7). Another possible explanation is

that SSA was varying at the end of the day as suggested in Section 4.2 and Figure 6. As observed for the single acquisition case, assuming clean snow provides a strong constraint and the correction is then excellent (Fig. 7). Although this strong constraint holds most of the time in winter, in spring at the end of the season, snow is rarely clean in mountains, and as discussed in Section 5.2, small slopes and small amounts of impurities can have similar signatures.

The correction method proposed by Dumont et al. (2017), applicable to time-series of albedo over dirty snow surfaces,

makes a stronger assumption to untangle the effect of slope and impurities. It assumes a known analytical formula for the shape of the spectrum, which applies when the type of impurities is perfectly known (e.g. BC or dust of known origin, or algae), and the absorption spectrum can be prescribed in the AART theory yielding a relationship between albedo, SSA and impurity concentrations. The joint optimisation of slope angles, SSA and impurity concentration is efficient as demonstrated at Col de Porte (Dumont et al., 2017) and Col du Lautaret (Tuzet, to be submitted). If these strong assumptions are acceptable,

and depending on the final goal of the correction, this method is certainly the most efficient.

Another important assumption used in the present paper is the perfect leveling of the sensors. It applies well to our measurements because the inclination angles are systematically recorded and proved to be stable within 0.2°. Because the effect of the tilt is of the same order as of the slope (in $K$, Eq. 18, the denominator is calculated with Eq. 2 using the tilt angles instead of the slope angles), this accuracy is sufficient. The method in Weiser et al. (2016) relaxes this assumption and the need for extra

measurements. It uses the diurnal cycle of reflected radiation instead of the albedo to infer the slope angles, while the sensor tilt is retrieved using the incident radiation cycle.

The wide range of possible assumptions shows that many methods can be valuable depending on the conditions. Further work should perform a more systematic comparison and exploration of the sensitivity to input uncertainties of each method.

## 6 Conclusions

Spectral albedo measured with horizontal sensors is very sensitive to the slope of the underlying surface in clear sky conditions, first because of the increased or decreased illumination received by the slope compared to a flat surface, and second because of additional illumination affecting the upward and downward looking sensors coming from the slope itself and the neighbouring slopes.



The first cause dominates up to about 15°("small slopes"), and has a detectable impact even for nearly flat surfaces, with 1-2° inclination. The main impact is a distortion of the spectrum shape embodied by a curvature in the visible range. For slopes facing the sun, the curvature is concave, peaking around 600 nm, which may result in an albedo value higher than 1. Nevertheless, even if less noticeable, the full spectral range and other slope configurations are affected by the slope effect. The

5      second cause becomes significant for slopes larger than about 15°. The theory for large slopes is analytically tractable in several particular cases but is more complex than for the small slopes and requires information or assumptions on the neighbouring slope, which limits its interest in practice. In all cases, the distortion due to the slope may greatly impact the calculation of the surface energy budget and snow properties retrievals (SSA, impurities) if the measured albedo is directly used in theories established for horizontal surfaces only.

10     The four spectral albedo correction methods proposed here for small slopes complement other methods presented in the literature for both spectral or broadband albedo. The diversity of methods is explained by the different possible assumptions that apply or not depending on the type of available measurements. More methods can be devised in the future, and the rigorous equation set provided in this paper should be helpful to this aim. Nevertheless, even though our results show that a satisfactory correction can be achieved in many situations (residual error better than 0.03), we emphasise that ancillary information is

15     required to perform such a correction, implying higher complexity and cost of instruments and protocols. In this context, our main recommendation is that slope inclination and aspect, and the diffuse-to-total irradiance ratio should be systematically recorded in future albedo measurement campaigns.

*Code and data availability.* The theory presented in Section 2 is implemented in the snowoptics library available from https://github.com/ghislainp/snowoptics and a webapp to interactively explore the slope effect is available here http://snowslope.pythonanywhere.com/.





## Appendix A: Apparent albedo for large slopes

### A1 Top-hill measurements and dark neighbourhood (case DT)

Combining Eqs. 14 and 16 and using $E_\lambda^{\mathrm{neigh,D}} = 0$ yields:

$$\acute{\alpha}_\lambda^{\mathrm{D,T}}(\theta_i) = \frac{V\left(\bar{\alpha}_\lambda^{\mathrm{dir}}(\acute{\theta}_i)E_\lambda^{\mathrm{sun}}\cos\acute{\theta}_i + \bar{\alpha}_\lambda^{\mathrm{diff}}V E_\lambda^{\mathrm{sky}}\right)}{E_\lambda^{\mathrm{sun}}\cos\theta_i + E_\lambda^{\mathrm{sky}}} \tag{A1}$$

The measured ratio between the diffuse flux obtained by obstructing the sun and the total received is equal to $r_\lambda$ in the case
of top-hill measurements. It follows that:

$$\acute{\alpha}_\lambda^{\mathrm{D,T}}(\theta_i) = (1 - r_\lambda)V K(\theta_i, \acute{\theta}_i)\bar{\alpha}_\lambda^{\mathrm{dir}}(\acute{\theta}_i) + r_\lambda V^2 \bar{\alpha}_\lambda^{\mathrm{diff}} \tag{A2}$$

This equation is similar to that for a small slope, except that the direct term is scaled by $V$ – due to screening of the incoming
diffuse radiation by the slope – while the diffuse term is scaled by $V^2$ – due to the combined effect of screened incoming diffuse
radiation and a reduced view of the slope by the downward looking sensor.

### A2 Mid-slope measurements and dark neighbourhood (case DM)

At mid-slope, the albedo has a different expression:

$$\acute{\alpha}_\lambda^{\mathrm{D,M}}(\theta_i) = \frac{V\left(\bar{\alpha}_\lambda^{\mathrm{dir}}(\acute{\theta}_i)E_\lambda^{\mathrm{sun}}\cos\acute{\theta}_i + \bar{\alpha}_\lambda^{\mathrm{diff}}V E_\lambda^{\mathrm{sky}}\right)}{E_\lambda^{\mathrm{sun}}S(\cos\acute{\theta}_i)\cos\theta_i + V E_\lambda^{\mathrm{sky}} + (1-V)\acute{E}_\lambda^{\uparrow}(\theta_i, \phi_i)} \tag{A3}$$

because it includes the light reflected by the portion of the slope above the sensor that illuminates the upward looking sensor.
This results in an additional term at the denominator compared to the case D, T (Eq. A3).

If the diffuse-to-total ratio is measured mid-slope by obstructing the sun, it is of the form:

$$\acute{r}_\lambda^{\mathrm{M}} = \frac{V E_\lambda^{\mathrm{sky}} + (1-V)\acute{E}_\lambda^{\uparrow}(\theta_i, \phi_i)}{E_\lambda^{\mathrm{sun}}S(\cos\acute{\theta}_i)\cos\theta_i + V E_\lambda^{\mathrm{sky}} + (1-V)\acute{E}_\lambda^{\uparrow}(\theta_i, \phi_i)} \tag{A4}$$

from which we can deduce the expression for the measured albedo:

$$\acute{\alpha}_\lambda^{\mathrm{D,M}}(\theta_i) = (1 - \acute{r}_\lambda^{\mathrm{M}})V K(\theta_i, \acute{\theta}_i)\bar{\alpha}_\lambda^{\mathrm{dir}}(\acute{\theta}_i) + \acute{r}_\lambda^{\mathrm{corr,D,M}}V\bar{\alpha}_\lambda^{\mathrm{diff}} \tag{A5}$$

This equation differs from Eq A2 for case T in two ways: first $V$ is not squared in the diffuse term and second the measured
diffuse-to-total ratio needs to be corrected from the illumination received by the upper slope. The measured ratio $r_\lambda^{\mathrm{corr,D,M}}$ can
be corrected with:

$$\acute{r}_\lambda^{\mathrm{corr,D,M}} = \acute{r}_\lambda^{\mathrm{M}} - \frac{(1-V)\acute{E}_\lambda^{\uparrow}(\theta_i, \phi_i)}{E_\lambda^{\mathrm{sun}}S(\cos\acute{\theta}_i)\cos\theta_i + V E_\lambda^{\mathrm{sky}} + (1-V)\acute{E}_\lambda^{\uparrow}(\theta_i, \phi_i)} \tag{A6}$$





and by replacing $\acute{\bar{E}}_\lambda^\uparrow(\theta_i, \phi_i)$ defined in Eq. 9:

$$\acute{r}_\lambda^{\text{corr,D,M}} = \acute{r}_\lambda^{\text{M}} - \frac{1-V}{V} \acute{\alpha}_\lambda^{\text{D,M}}(\theta_i). \tag{A7}$$

This equation is interesting *per se* to get the diffuse-to-total ratio above the slope ($r_\lambda$) from the measured diffuse-to-total ratio. However, it can also be combined with Eq. A5 to yield a closed form for $\acute{\alpha}^{\text{D,M}}$:

$$5 \quad \acute{\alpha}_\lambda^{\text{D,M}}(\theta_i) = (1 - \acute{r}_\lambda^{\text{M}}) \frac{V}{1+M_\lambda} K(\theta_i, \acute{\theta}_i) \bar{\alpha}_\lambda^{\text{dir}}(\acute{\theta}_i) + \acute{r}_\lambda^{\text{M}} \frac{V}{1+M_\lambda} \bar{\alpha}_\lambda^{\text{diff}}. \tag{A8}$$

Another useful derivation is when the diffuse-to-total ratio is modelled using an atmospheric model, i.e. similar to when measured at the top of the hill. The ratio is then defined as in Eq. 19 and the albedo can be expressed as:

$$\acute{\alpha}_\lambda^{\text{D,M}}(\theta_i) = \frac{\acute{\alpha}_\lambda^{\text{D,T}}(\theta_i)}{1 - (1-V)r_\lambda + \frac{1-V}{V} \acute{\alpha}_\lambda^{\text{D,T}}(\theta_i)} \quad \text{if } S(\cos\acute{\theta}_i) > 0. \tag{A9}$$

When the sun disappears below the slope, the previous equation is not valid anymore because the upward looking sensor records a discontinuous drop of irradiance, resulting in a different formulation for the albedo that we straightly derive from Eqs. 9 and A3:

$$\acute{\alpha}_\lambda^{\text{D,M}}(\theta_i) = \frac{V \bar{\alpha}_\lambda^{\text{diff}}}{1+M_\lambda} \quad \text{if } S(\cos\acute{\theta}_i) = 0 \tag{A10}$$

**A3   Top-hill measurements and snow-covered neighbourhood (case ST)**

The energy received by the downward looking sensor in the case of snow is obtained by injecting Eqs 12 and 13 into Eq. 14:

$$15 \quad I_\lambda^{\text{d,S}}(\theta_i, \phi_i) = \frac{\left((V + M_\lambda(1-V))\bar{\alpha}_\lambda^{\text{dir}}(\acute{\theta}_i)\cos\acute{\theta}_i + (M_\lambda V + (1-V))\bar{\alpha}_\lambda^{\text{dir}}(\theta_i)\cos\theta_i\right)E_\lambda^{\text{sun}}}{1 - M_\lambda^2} + \frac{\bar{\alpha}_\lambda^{\text{diff}} V E_\lambda^{\text{sky}}}{1 - M_\lambda}. \tag{A11}$$

Dividing by the incoming irradiance at the top of the slope $E_\lambda^{\text{sun}}\cos\theta_i + E_\lambda^{\text{sky}}$, gives the albedo measured near the top of the slope when the area is fully covered by snow:

$$\acute{\alpha}^{\text{S,T}} = (1 - r_\lambda)\left[\frac{V + M_\lambda(1-V)}{1 - M_\lambda^2} K(\theta_i, \acute{\theta}_i)\bar{\alpha}_\lambda^{\text{dir}}(\acute{\theta}_i) + \frac{M_\lambda V + (1-V)}{1 - M_\lambda^2} \bar{\alpha}_\lambda^{\text{dir}}(\theta_i)\right] + r_\lambda \frac{V}{1 - M_\lambda} \bar{\alpha}_\lambda^{\text{diff}} \tag{A12}$$

**A4   Mid-slope measurements and snow-cover neighbourhood (case SM)**

The energy received by the downward and upward looking sensors is given by Eqs. A11 and 15 respectively leading to the albedo:

$$\acute{\alpha}_\lambda^{\text{S,M}}(\theta_i) = (1 - \acute{r}_\lambda^{\text{M}})\left[\frac{V + M_\lambda(1-V)}{1 - M_\lambda^2} K(\theta_i, \acute{\theta}_i)\bar{\alpha}_\lambda^{\text{dir}}(\acute{\theta}_i) + \frac{M_\lambda V + (1-V)}{1 - M_\lambda^2} \bar{\alpha}_\lambda^{\text{dir}}(\theta_i)\right] \tag{A13}$$

$$+ \acute{r}_\lambda^{\text{corr,S,M}} \frac{1}{1 - M_\lambda} \bar{\alpha}_\lambda^{\text{diff}} \quad \text{if } S(\cos\acute{\theta}_i) > 0. \tag{A14}$$





where $r_\lambda^{\mathrm{corr,S,M}}$ is given by Eq. A6. Unfortunately it is not possible to obtain a simple form as in the D, M case (Eq. A7) because we use the fact that for a dark neighbourhood the downwelling irradiance from the upper slope reaching the upward looking sensor is equal to the upwelling flux from the slope reaching the downward looking sensor. The neighbourhood contribution in the snowy case on the flux reaching the downward looking sensor voids this simple relationship. Nevertheless, it is possible to inject the expression for $\acute{E}_\lambda^{\uparrow,S}(\theta_i, \phi_i)$ for snow (Eq. 12) into Eq. A6, leading to:

$$\acute{r}_\lambda^{\mathrm{corr,M}} = \acute{r}_\lambda^{\mathrm{M}} - \frac{1-V}{1-M_\lambda^2}\left(\bar{\alpha}_\lambda^{\mathrm{dir}}(\acute{\theta}_i)K(\theta_i,\acute{\theta}_i) + \bar{\alpha}_\lambda^{\mathrm{dir}}(\theta_i)M_\lambda\right)(1-\acute{r}_\lambda^{\mathrm{M}}) - \frac{1-V}{1-M_\lambda}\bar{\alpha}_\lambda^{\mathrm{diff}}\acute{r}_\lambda^{\mathrm{corr,M}} \tag{A15}$$

and when solving for $\acute{r}_\lambda^{\mathrm{corr,M}}$:

$$\acute{\alpha}_\lambda^{\mathrm{S,M}}(\theta_i) = (1-\acute{r}_\lambda^{\mathrm{M}})\left[\frac{V}{1+M_\lambda}K(\theta_i,\acute{\theta}_i)\bar{\alpha}_\lambda^{\mathrm{dir}}(\acute{\theta}_i) + \frac{1-V+M_\lambda}{1+M_\lambda}\bar{\alpha}_\lambda^{\mathrm{dir}}(\theta_i)\right] + \acute{r}_\lambda^{\mathrm{M}}\bar{\alpha}_\lambda^{\mathrm{diff}} \text{ if } S(\cos\acute{\theta}_i) > 0. \tag{A16}$$

The expression of the albedo when the diffuse-to-total ratio is available from atmospheric modelling far above the topography is derived following Eq. A9:

$$\acute{\alpha}_\lambda^{\mathrm{S,M}}(\theta_i) = \acute{\alpha}_\lambda^{\mathrm{S,T}}(\theta_i)\frac{1}{1-(1-V)r_\lambda + \frac{(1-V)\acute{E}_\lambda^{\uparrow}(\theta_i,\phi_i)}{E_\lambda^{\mathrm{sun}}\cos\theta_i + E_\lambda^{\mathrm{sky}}}}. \tag{A17}$$

Because of the neighbourhood contribution, the term $\acute{E}_\lambda^{\uparrow}(\theta_i, \phi_i)$ can not be simply related to the albedo as it was the case for the dark case (Eq. A9). A fully developed expression can be nevertheless obtained by injecting Eq 12:

$$\acute{\alpha}_\lambda^{\mathrm{S,M}}(\theta_i) = \frac{\acute{\alpha}_\lambda^{\mathrm{S,T}}(\theta_i)}{1+(1-V)\left[(1-r_\lambda)\left(\frac{1}{1-M_\lambda^2}K(\theta_i,\acute{\theta}_i)\bar{\alpha}_\lambda^{\mathrm{dir}}(\acute{\theta}_i) + \frac{M_\lambda}{1-M_\lambda^2}\bar{\alpha}_\lambda^{\mathrm{dir}}(\theta_i)\right) + r_\lambda\left(\frac{V}{1-M_\lambda}\bar{\alpha}_\lambda^{\mathrm{diff}}-1\right)\right]} \tag{A18}$$

When the sun is below the slope, we shall distinguish two cases, first when the upward looking sensor is shadowed, but the neighbouring surface is still illuminated, and second when the neighbouring surface is also in the shadows. The first case is not tractable because $E^{\mathrm{sun}}$ is not recorded by any of the sensors, whereas the neighbourhood term depends on this term. It is therefore impossible to provide an expression of the measured albedo without expliciting $E^{\mathrm{sun}}$. The second case, when the whole area is in the shadow is obtained by noting that $\acute{E}_\lambda^{\uparrow,S}(\theta_i, \phi_i) = E_\lambda^{\mathrm{neigh,S}} = \frac{1}{1-M_\lambda}\bar{\alpha}_\lambda^{\mathrm{diff}}VE_\lambda^{\mathrm{sky}}$ and yields a trivial result:

$$\acute{\alpha}_\lambda^{\mathrm{S,M}}(\theta_i) = \bar{\alpha}_\lambda^{\mathrm{diff}} \text{ if } S(\cos\acute{\theta}_i) = 0. \tag{A19}$$

This expression is in fact included in Eq. A16 when $\acute{r}_\lambda^{\mathrm{M}} = 1$.



*Author contributions.* G.P. and M.D. designed the study and performed the theoretical calculations. M.L., F.T., F. L, and L.A contributed to the measurements. All authors contributed to the reflection on the slope effect and to the manuscript.

*Competing interests.* The authors declare no competing interests.

*Acknowledgements.* The devices to measure spectral albedo were developed under the ANR program 1-JS56-005-01 MONISNOW and
5  by a grant from OSUG@2020 (investissement d'avenir – ANR10 LABX56). Many measurements were collected in the framework of the ANR program ANR-16-CEO01-0006 EBONI and LEFE ASSURANCE. We also acknowledge the French Spatial Agency (CNES TOSCA Miosotis) for the financial support and the Academy of Finland (contract number 304345). Alexander Kokhanovky provided very helpful comments on the manuscript.





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


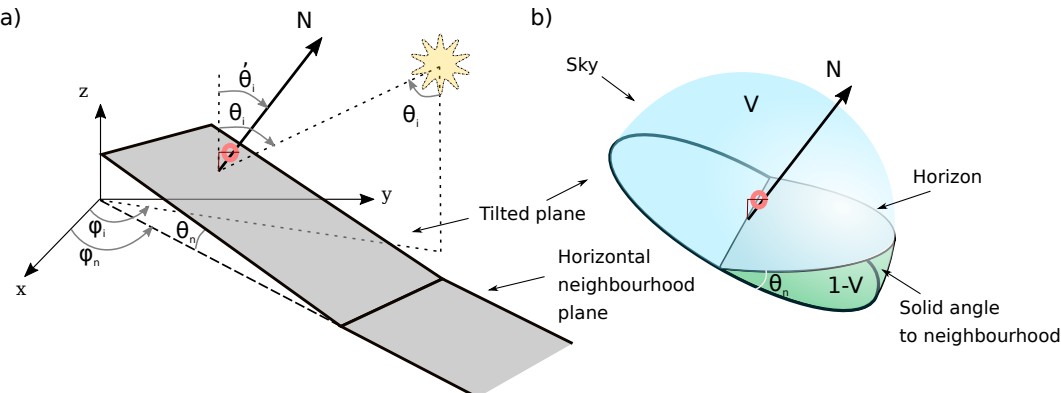

**Figure 1.** a) Geometry of the sloped infinite plane illuminated by the sun. The red symbol represents the upward and downward looking sensors. b) Sky solid angle (blue shade) used to compute $V$, and neighbourhood solid angle (green).

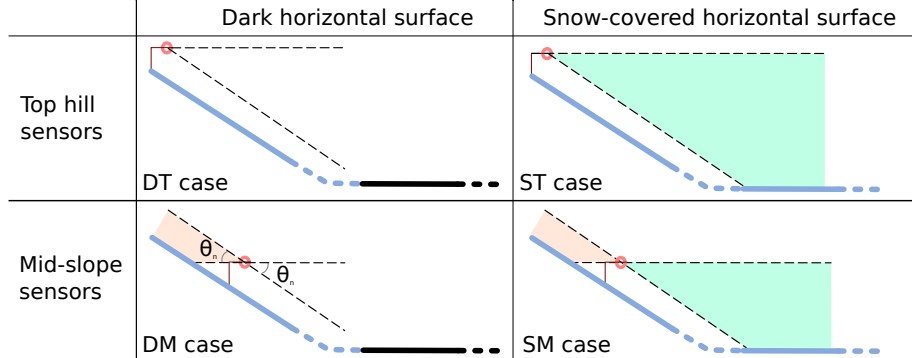

**Figure 2.** The four studied cases depending on the sensor position (mid-slope, top hill) and cover type of the horizontal surface: snow (blue), dark (black). The neighbourhood (i.e. horizontal surface) is seen by the downward-looking sensor in the lower solid angle (green shade, as in Fig. 1). The upper slope is seen by the upward-looking sensor by the upper solid angle (orange shade). Note that the horizontal surface is infinite and the slope is infinite except in the top-hill case.



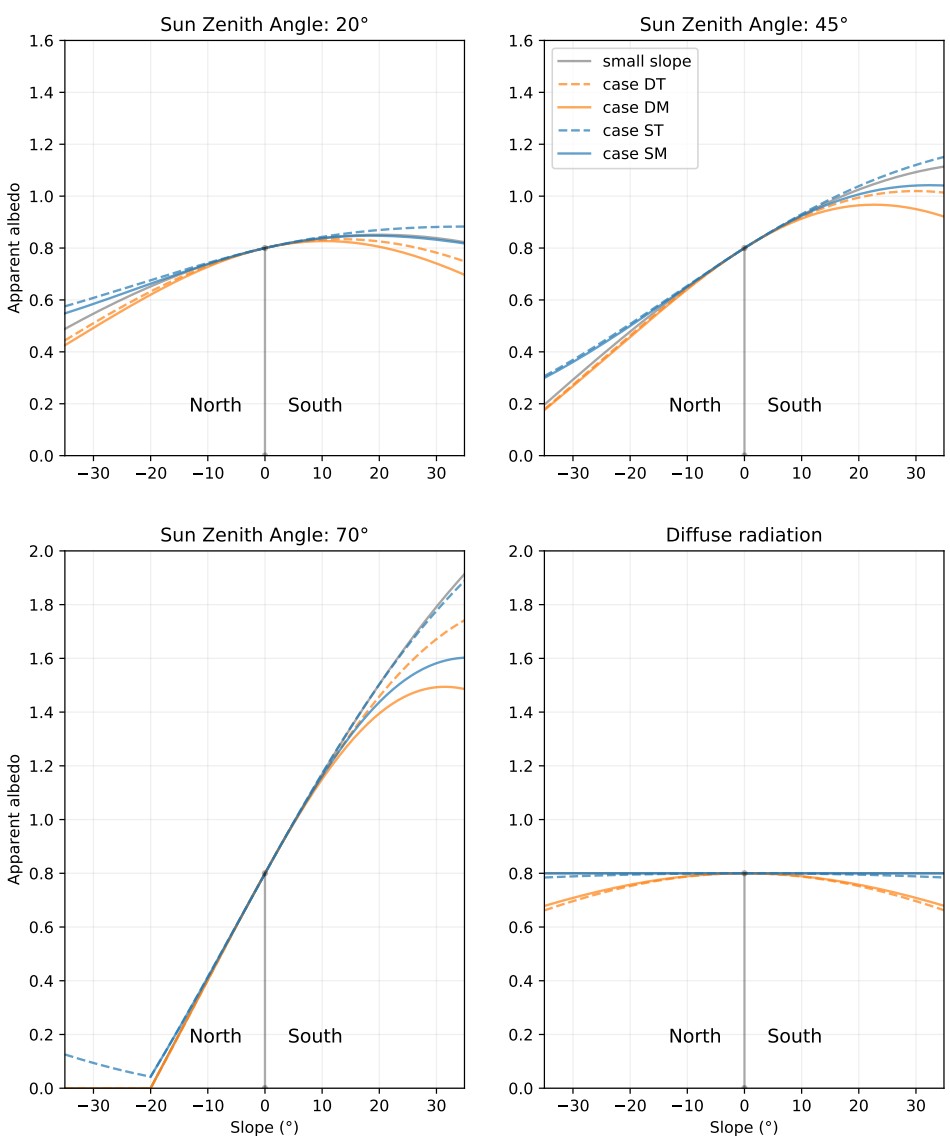

**Figure 3.** Apparent albedo as a function of slope computed for various formulation and different illumination conditions. The flat albedo is here fixed to 0.8 and the no diffuse radiation is considered.

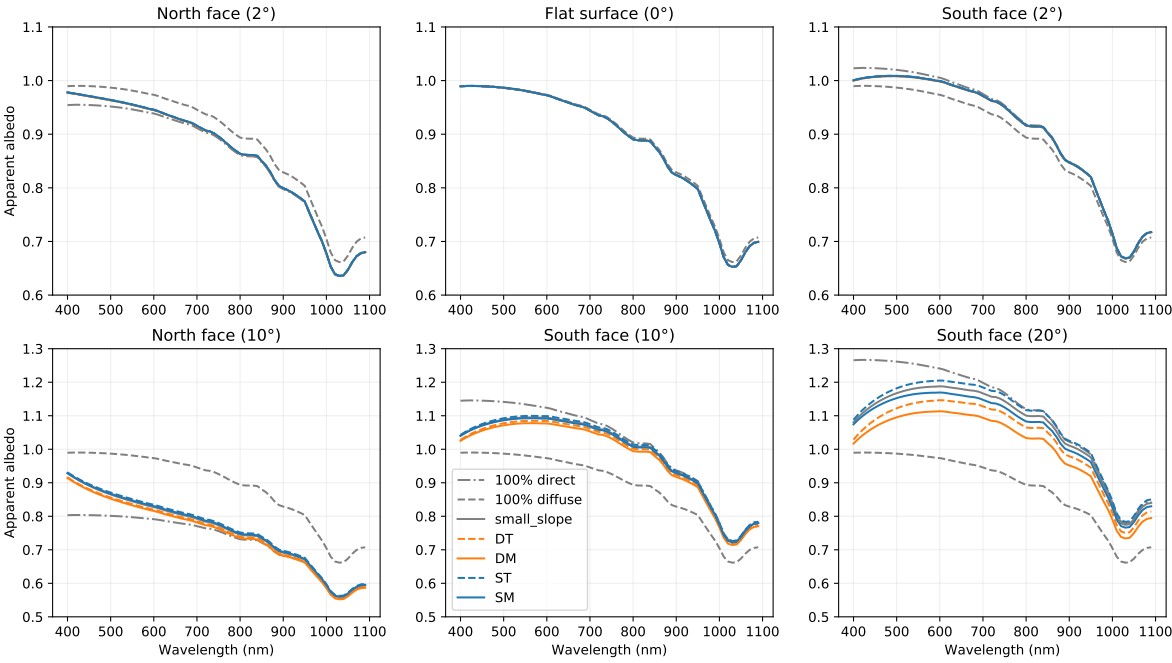

**Figure 4.** Spectra of apparent albedo for various slopes under blue-sky conditions (the diffuse irradiance decreases as a power 4 of the wavelength). The grey curves are calculated with the small slope approximation.





**Figure 5.** Measured, calculated and corrected albedo for 7 acquisitions (rows) taken in different terrain configurations. The first column compares apparent albedo calculated from the theory (small slope and SM case) to measured albedo. The second column compares corrected albedo using measured slope parameters to intrinsic diffuse albedo calculated for a flat surface using measured SSA (gray). The third column is similar to the second column, except that the measured albedo is corrected without using measured slope parameters but by assuming clean snow. The second and third columns also show measured albedo to highlight the change due to the correction.



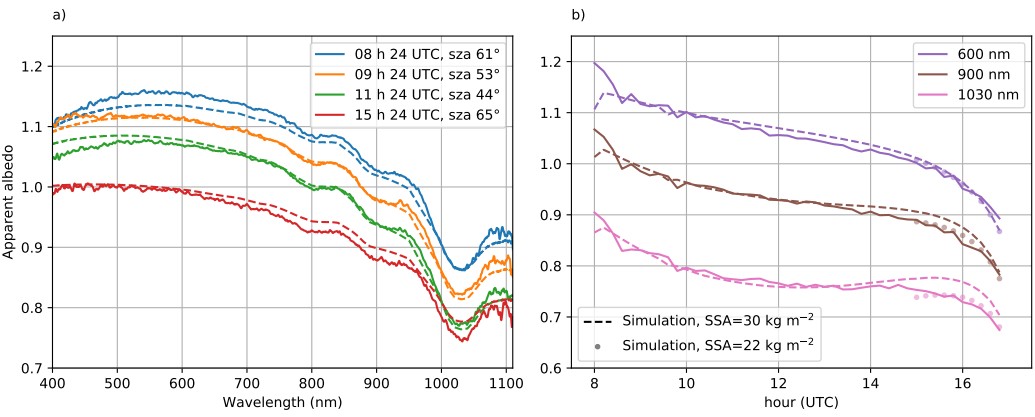

**Figure 6.** Measured (plain) and simulated (dashed) albedo acquired on 23 March 2018. a) albedo spectra acquired at four selected hours during the day, b) albedo as a function of time for three selected wavelengths.

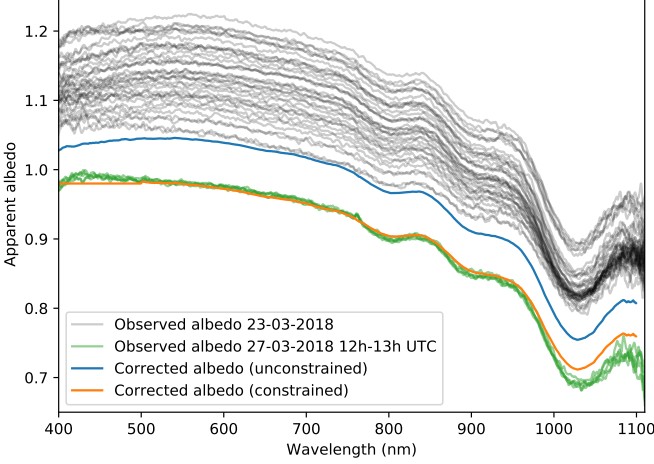

**Figure 7.** Albedo spectra measured with Autosolexs (gray) and corrected (orange, blue) on 23 March 2018. Measured albedo during overcast conditions on 2 March 2018 are also shown (green).



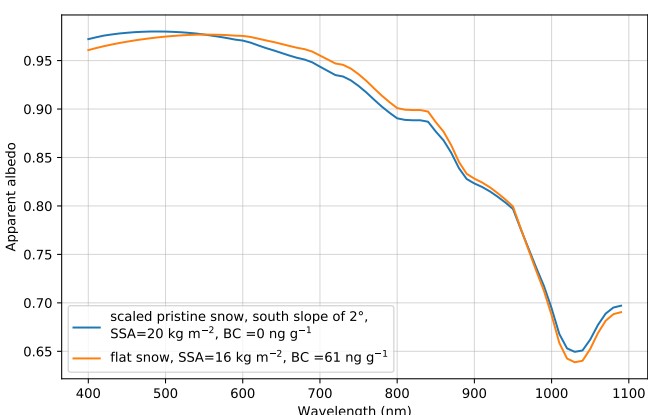

**Figure 8.** Example of scaled spectrum of pristine snow affected by a small slope (2°) and comparison to dirty snow spectrum with fitted SSA and black carbon (BC) content.