# Peer review of "Spectral albedo measurements over snow-covered slopes: theory and slope effect corrections"

_The Cryosphere, 2019_

## Referee Comment (RC1) · Anonymous Referee #1 · 8 Feb 2020

This manuscript investigates the effects of slope on snow albedo measurements both theoretically and practically and provides different correction methods based on auxiliary data availability. The analysis is comprehensive. The flow is coherent. The language is concise. First, the introduction provides adequate background knowledge to inform and intrigue readers, including the importance of the topic, previous practices to mitigate this bias in measured snow albedo, and the potential obstacles. Next, the theoretical analysis is thorough and clear. The authors break down this complicated problem into well-considered aspects in a well-paced manner. The mathematical forms are clear. So are their physical meanings. Last, the application is practical. The authors elaborate when and how to apply their methods and discuss in-depth the caveats caused by some of the assumptions. Researchers working on snow albedo and surface energy budget would benefit greatly from this manuscript. I suggest "minor revision". Please see my comments below.

1, Page 1 Line 5: "Here we investigate…"
At this point, the significance of the topic is not clear yet. The statement of "what we did" would appear immature. Like what Dr. Joshua Schimel said, "you try to give me a solution when I do not know I have a problem". Since broad vs spectral albedo is not the No.1 priority of this manuscript, I suggest emphasizing the effects of slope on albedo in the first few sentences instead.

2, Page 2 Line 13: Section 5 → Section 5 and 6

3, Page 2 Line 23, Both → Both surfaces

4, Figure 1a, correct me if I am wrong, should $\theta'_i \rightarrow \theta^n$. Please see my drawing in the figure below

[Figure]

5, Figure 1b, I suggest making the periphery of the horizontal hemisphere as dashed lines to prevent from confusing with "horizontal neighborhood plane".

6, Section 2.2, will sensor viewing angles make a difference in upwelling and downwelling radiation on sensors (eg, 160° rather than 180°)?

7, Page 7 Line 8, A first case → The first case

8, Page 8 Line 17, INSERT TABLE 1 HERE. I believe these will be deleted in the final version?

9, Table 1: Please highlight the first row and column (including the dividers) to make it clearer.

10, Page 11 Line 7, would you please inform us during which months these data were measured?

11, Page 11 Line 9/20, Solab or Solalb?

12, Figure 3
   1) Legend missing
   2) Please use a different line style to separate North and South (different from the "small slope")
   3) y-axis, how about using "albedo change by slope" (Apparent albedo minus 0.8) which will make the effects of slopes clearer?
   4) I suggest moving panel titles to left-aligned or inside the panels. Otherwise, titles of the second row look like x-axis labels of the first row.

13, Page 12 Line 8-9, "Neglecting …"
This sentence involves too many numbers. Please break it down. One way to do it is to summarize the idea here (a revised form of the next sentence would do) and mention these numbers as you describe each SZA below.

14, Page 12 Line 19,

15, Page 12 Line 19, lowest → smallest

16, Page 12 Line 20-22, the active sense and the use of "observed" and "measurements" make it confusing whether this is theoretical analysis or not.
Suggestion: Incoming radiation at the downward-looking sensor has a deficit …
Use "estimate" or "calculate" instead of "observe" and "measure" here.

17, Page 12 Line 25, how about a figure same as Figure 3 but for diffuse radiation in the supplement? Or a more quantitative description here.

18, Page 13 Line 32, Fig. 4 → Fig. 4)  (missing the right parenthesis)

19, Page 14 Line 18, simulated → calculated

Please keep it consistent. There are already enough types of albedo here.

20, Page 15 Line 15, please break down this long sentence.

21, Page 15 Line 23-25, "The correction method …" → "This method yields better results than that with the measured slope parameters"

22, Page 15 Line 26, suspect → suspicious? spurious? false?
Or state the problem directly, eg, too flat?

23, Page 19 Line 32, increased → excessive; decreased → deficient

23, Page 19 Line 32, "additional illumination …" → "the upward and downward-looking sensors affected by additional illumination coming from ..."
"Illumination" and "coming from" were too far away.

---

## Referee Comment (RC2) · Anonymous Referee #2 · 24 Mar 2020

General assessment ===

In this manuscript, Picard et al. describe various correction procedures for broadband and spectral albedo measurements performed over sloping surfaces. Out of many possible assumptions and simplifications, they opt for a theoretical treatment differentiated by surface conditions of the horizontal far-field (dark or snow-covered) and location of the observation platform (mid-slope or near the top of the slope). This yields a matrix of 4 possible correction procedures.

Several practical recipes are provided for slope corrections, depending on the available information (e.g. slope parameters, time series, etc.)

[Figure]

The combination of rigorous theoretical approach and practical applicability makes this a very useful and accessible text for anyone involved in observations of (spectral) albedo in sloping terrain.

I recommend publication of this article in The Cryosphere following only textual improvements detailed below.

===

P1L4 properties -> processes

P1L8 appointment -> partitioning

P1L20 barely invisible -> barely visible or almost invisible

P2L5 such as -> such that

P2L7: an -> the

P2L20: I understand that this is a sensitive remark, but Pirazzini (2004) is an appropriate example of a publication where slope effects are mistakenly interpreted as a diurnal change of snow properties, so this could be cited here.

P2L25 insert comma after "Similarly"

P2L30 Suggest "Such an accuracy cannot be achieved in practice, because the measurement ...."

P3L11: algea -> algae

P4L3: I understand what you want to say here, but as it is formulated, it is not correct. Suggest "and to this end, analytical formulations using simplifying assumptions are preferred at the cost of model complexity."

P5 and further: I would like you to make sure that you mean to write É instead of E' (similarly with theta ′ instead of theta')

[Figure]

P6L2: titled -> tilted

P6L11: an horizontal -> a horizontal

P6L11 and L13: cover -> surface type

P7L15: slightly slopes -> small slopes or gentle slopes

P7L20: perhaps this is a good place to reiterate once more the meaning of the abbreviations: D and S for dark and snow, and M and T for mid-slope and top measurements.

P8L19: moderately -> moderate

P12L25: Suggest "The slope effect is largest under direct illumination, which occurs most in the near-infrared domain under clear-sky conditions."

P13L7: superficial -> near-surface (also P16L9)

P14L1: slopes -> slope

P14L19: at last -> lastly

P14L33: perhaps this is a good place to insert the possibility of drone-derived DEMs of small areas in order to compute surface slope in a non-invasive way.

P16L13: variable -> mixed

P18L20: systematically -> systematical

Figure 3: suggest to discriminate more between blue and gray

---

## Author Comment (AC1) · 1 Apr 2020

This manuscript investigates the effects of slope on snow albedo measurements both theoretically and practically and provides different correction methods based on auxiliary data availability. The analysis is comprehensive. The flow is coherent. The language is concise. First, the introduction provides adequate background knowledge to inform and intrigue readers, including the importance of the topic, previous practices to mitigate this bias in measured snow albedo, and the potential obstacles. Next, the theoretical analysis is thorough and clear. The authors break down this complicated problem into well-considered aspects in a well-paced manner. The mathematical forms are clear. So are their physical meanings. Last, the application is practical. The authors elaborate when and how to apply their methods and discuss in-depth the caveats caused by some of the assumptions. Researchers working on snow albedo and surface energy budget would benefit greatly from this manuscript. I suggest "minor revision".

We thank the reviewer for these general, supportive, comments. We have taken  into account all the detailed corrections listed below, almost as proposed.

Please see my comments below.

1, Page 1 Line 5: "Here we investigate..."
At this point, the significance of the topic is not clear yet. The statement of "what we did" would appear immature. Like what Dr. Joshua Schimel said, "you try to give me a solution when I do not know I have a problem". Since broad vs spectral albedo is not the No.1 priority of this manuscript, I suggest emphasizing the effects of slope on albedo in the first few sentences instead.

The beginning of the abstract is reformulated in a more symmetrical way. However, we kept the mention of  spectral albedo in the objective: "Here we investigate the sensitivity of spectral albedo measurements to surface slope", because even though the theory is not spectral, the corrections and the examples are most relevant for spectral only. The application of the correction to broadband albedo would require significant changes.

2, Page 2 Line 13: Section 5 → Section 5 and 6

done

3, Page 2 Line 23, Both → Both surfaces

Done

4, Figure 1a, correct me if I am wrong, should θ , i → θ n . Please see my drawing in the figure below

The figure is indeed incorrect. It is now corrected.

5, Figure 1b, I suggest making the periphery of the horizontal hemisphere as dashed lines to prevent from confusing with "horizontal neighborhood plane".

Done

6, Section 2.2, will sensor viewing angles make a difference in upwelling and downwelling radiation on sensors (eg, 160 o rather than 180 o )?

The sensors not being perfect, i.e. they have a reduced field of view or they do not have a perfect cosine response, implies that an "instrumental response" function should be added and it would be very difficult to track the equations. We have added a sentence in the beginning of Section 2.2 to make our assumption of perfect sensors explicit: "The upward and downward looking sensors are considered to be horizontal and to have a perfect cosine response with a 180° field of view."

7, Page 7 Line 8, A first case → The first case

Done

8, Page 8 Line 17, INSERT TABLE 1 HERE. I believe these will be deleted in the final version?

Yes, this is to help the editor in positioning the table.

9, Table 1: Please highlight the first row and column (including the dividers) to make it clearer.

That would be better but we have followed the template provided by The Cryosphere. We will contact the editor to discuss how this can be improved.

10, Page 11 Line 7, would you please inform us during which months these data were measured?

The information is added.

11, Page 11 Line 9/20, Solab or Solalb?

Solalb, we have corrected.

12, Figure 3

1) Legend missing
2) Please use a different line style to separate North and South (different from the "small slope")
3) y-axis, how about using "albedo change by slope" (Apparent albedo minus 0.8) which will make the effects of slopes clearer?
4) I suggest moving panel titles to left-aligned or inside the panels. Otherwise, titles of the second row look like x-axis labels of the first row.

We have applied the suggested changes 2, 3 and 4. Regarding point 1, we understand "legend missing" as to repeat the legend box in the four panels. We have tried, but this results in an overload of the graphs for a weak benefit.

13, Page 12 Line 8-9, "Neglecting ..."
This sentence involves too many numbers. Please break it down. One way to do it is to summarize the idea here (a revised form of the next sentence would do) and mention these numbers as you describe each SZA below.

We have split the sentence.

14, Page 12 Line 19, remove "which is very large"
done.

15, Page 12 Line 19, lowest → smallest

16, Page 12 Line 20-22, the active sense and the use of "observed" and "measurements" make it confusing whether this is theoretical analysis or not.
Suggestion: Incoming radiation at the downward-looking sensor has a deficit ...
Use "estimate" or "calculate" instead of "observe" and "measure" here.

We have reformulated the two sentences, using "position" to indicate where the sensors is, thus avoiding the word "measurement". Observed is replaced by estimated.

17, Page 12 Line 25, how about a figure same as Figure 3 but for diffuse radiation in the supplement? Or a more quantitative description here.

The information is already in Fig 3. To point the reader to the figure again, we have added a reference to the bottom right panel, because it is true that the last reference to the figure was high above in the text.

18, Page 13 Line 32, Fig. 4 → Fig. 4) (missing the right parenthesis)

Done.

19, Page 14 Line 18, simulated → calculatedPlease keep it consistent. There are already enough types of albedo here.

Done.

20, Page 15 Line 15, please break down this long sentence.

Done.

21, Page 15 Line 23-25, "The correction method ..." → "This method yields better results than that with the measured slope parameters"

Done.

22, Page 15 Line 26, suspect → suspicious? spurious? false?
Or state the problem directly, eg, too flat?

We have use "seems too flat".

23, Page 19 Line 32, increased → excessive; decreased → deficient

We have reformulated using "change" since "excessive" or "deficient" are not neutral.

23, Page 19 Line 32, "additional illumination ..." → "the upward and downward-looking sensors affected by additional illumination coming from ..."
"Illumination" and "coming from" were too far away.

Done.

---

## Author Comment (AC2) · 1 Apr 2020

General assessment ===
In this manuscript, Picard et al. describe various correction procedures for broadband
and spectral albedo measurements performed over sloping surfaces. Out of many possible assumptions
and simplifications, they opt for a theoretical treatment differentiated by surface conditions of the
horizontal far-field (dark or snow-covered) and location of the observation platform (mid-slope or near
the top of the slope). This yields a matrix of 4 possible correction procedures. Several practical recipes
are provided for slope corrections, depending on the available information (e.g. slope parameters, time
series, etc.). The combination of rigorous theoretical approach and practical applicability makes this a
very useful and accessible text for anyone involved in observations of (spectral) albedo in sloping
terrain.
I recommend publication of this article in The Cryosphere following only textual improvements
detailed below.

We thank the reviewer for the general analysis of our paper and the detailed comments.

comment
===
P1L4 properties -> processes

We have removed this sentence based on RC1 comments. In any case, we do agree with the
proposition.

P1L8 appointment -> partitioning

Done.

P1L20 barely invisible -> barely visible or almost invisible

We have changed to "almost invisible".

P2L5 such as -> such that

Done.

P2L7: an -> the

Done.

P2L20: I understand that this is a sensitive remark, but Pirazzini (2004) is an appropriate example of a
publication where slope effects are mistakenly interpreted as a diurnal change of snow properties, so
this could be cited here.

The short answer: It is a legitimate and interesting remark in an open discussion journal. For the final
version of the scientific paper, however, we can only claim that there is a mistake if we can prove it.
This would require to reanalyze the data, and most probably to know the actual slope under the sensor,
which is not possible anymore. Furthermore the 2004 paper was about broadband, which is not cover in
the present paper. The lack of direct/diffuse partitioning would prevent anyway to conclude.

 The long answer by the author of the study (Pirazzini, 2004):

The interpretation of the diurnal cycle of broadband albedo in the 2004 paper was given after ruling out other possible explanations, starting from the slope of the surface. The surfaces under the sensors located at Newmayer and Hells Gate were as much horizontal as a natural surface can be. After the publication of the paper, AWI colleagues were surprised by my results and analysed other data from Neumayer. They have visited the site several times, and confirmed that there are no persistent sastrugi, and that the diurnal behaviour that I saw in two selected clear-sky days is common also in periods that I did not analyse. They finally agreed on (or did not oppose to) my conclusions that the most reasonable explanation for the observed diurnal cycle is that snow metamorphism dominated over the solar zenith angle effect in summer.

In the case of Dome C I had less data and less information on the site, so my interpretation was based on the visual observations made by the colleagues from CNR who collected the measurements, and on the similarity with the diurnal cycle observed also in other snow covered Arctic sites (not presented in the paper).

Anyhow, whatever is the case for Dome C, the conclusions in the paper hold for Newmayer and Hells Gate, and the effect of sastrugi is demonstrated through the observations taken at Reeves Névé before and after the rotation of the horizontal arm holding the pyranometers. So, I consider the present reference to the 2004 paper very appropriate: "the local slope in the footprint can be significant because of sastrugi or dunes (Grenfell et al., 1994; Warren et al., 1998; Pirazzini, 2004; Wang and Zender, 2010)", but I disagree with using this reference for the preceeding sentence.

P2L25 insert comma after "Similarly"

Done.

P2L30 Suggest "Such an accuracy cannot be achieved in practice, because the measurement .…"

Done.

P3L11: algea -> algae

Done.

P4L3: I understand what you want to say here, but as it is formulated, it is not correct.
Suggest "and to this end, analytical formulations using simplifying assumptions are preferred at the cost of model complexity."

We have changed to " and to this end, analytical formulations using simplifying assumptions are preferred over complex models".

P5 and further: I would like you to make sure that you mean to write É instead of E'(similarly with theta 0 instead of theta')

Yes. Although it is an unusual symbol, it represents a tilted terrain. A longer line, with constant width would be better but does not exist.

P6L2: titled -> tilted

Done.

P6L11: an horizontal -> a horizontal

Done.

P6L11 and L13: cover -> surface type

Done.

P7L15: slightly slopes -> small slopes or gentle slopes

We have changed to "Small slopes".

P7L20: perhaps this is a good place to reiterate once more the meaning of the abbreviations: D and S for dark and snow, and M and T for mid-slope and top measurements.

Done.

P8L19: moderately -> moderate
P12L25: Suggest "The slope effect is largest under direct illumination, which occurs most in the near-infrared domain under clear-sky conditions."

Corrected, as proposed.

P13L7: superficial -> near-surface (also P16L9)

Done.

P14L1: slopes -> slope

Done.

P14L19: at last -> lastly

Done.

P14L33: perhaps this is a good place to insert the possibility of

We have added: "UAV and laser scanners are possible tools to acquire accurate snow digital surface model in a non-invasive way."

P16L13: variable -> mixed

Done.

P18L20: systematically -> systematical

We have completely removed the word.

Figure 3: suggest to discriminate more between blue and gray

We have added crosses for the small slope calculation.